

# How to communicate and educate more effectively on natural risk issues to improve disaster risk management through serious games

Mercedes Vázquez-Vílchez[1*], Rocío Carmona-Molero[1,2], Tania Ouariachi-Peralta[3]

[1]Didactics of Experimental Sciences, University of Granada, Granada, 18011 Granada, Spain.
[2]Department of Analytical Chemistry, University of Granada, Granada, 18071 Granada, Spain.
[3]Professorship Communication, Behaviour & The Sustainable Society, Center of Expertise Energy, Hanze University of Applied Sciences, Groningen, 9747 AS, The Netherlands.

*Correspondence to*: Mercedes Vázquez-Vílchez (mmvazquez@ugr.es)

**Abstract.** This study focuses on exploring the potential of serious games for improving disaster risk management. The research involves methodological triangulation, analysing and comparing data from content analysis of serious games (6 digital games: 3 mobile apps and 3 online games), focus groups with experts and literature review. The results show that only online games fulfil the fundamental narrative indicated by the experts, with mobile apps focusing their gameplay more on interaction. Such interaction could enhance the playful aspect of the game and thus increase the desire to play; thus, the educational aspect of online games is much higher. Few online games work on issues of multiculturalism, diversity and gender. This paper provides a list of recommended features of disaster risk management games that we have categorised into three dimensions: a) character, b) information and message tone and c) narrative dynamics, reward systems and feedback. The results can be of great help to teachers and game designers in improving citizens' knowledge of disaster risk management.

## 1 Introduction

Today's scientific and technological advances allow us to anticipate natural hazards and take early action, both at governmental and civilian levels. However, the occurrence of devastating disasters in countries of any economic and cultural scale shows that these technological and scientific advances in disaster risk management (DRM) do not necessarily correspond to their correct implementation. Examples are the catastrophic floods that occurred in Europe (Germany, Belgium and the Netherlands) in 2021 or the earthquake that affected Turkey and Syria in 2023, where many people died and economic losses were very high. Therefore, there is a huge gap between scientific-technological valuations and their practices and implementation, and this communication is a critical factor in DRM (Solinska-Nowak et al., 2018; Weyrich et al., 2021).

The dynamics of conflict resolution caused by natural hazards-related disaster are based on centralized processes in which decision-making is linked to governments, scientists and experts (Clerveaux et al., 2008; Tanwattana and Toyoda, 2018)



which minimizes the participation of affected communities. In extreme cases, these decisions may even be made without regard to the local cultural, social or economic norms of the affected area. In this context, in the 21st century, there has been a growing interest in changing such hierarchical decision-making and converting it into more participatory strategies involving communities (Yamori and Kikkawa, 2005; Yamori, 2007; Yamori, 2008; Suarez, et al., 2014; Tanwattana and

Toyoda, 2018). With this point of view, society is not understood as a world where there is a need for an only one solution proposed by a people such as scientists or politicians, but as a society where the dialogue is possible and diverse viable answers coexist (Yamori, 2011). Some authors suggested a mutual learning, in order to promote the democratization of decisions, where combines diverse learning methodology, such as adaptive management, experiential learning, or transformative learning (e.g. Lavell et al., 2012).

Following is approach, in which acquiring knowledge about natural hazards should enable citizens to make decisions and implement prevention measures, there is recognition of active teaching methodologies, such as serious games, which may serve as a participatory and supportive tool for understanding the essential aspects of natural hazards (e.g. Solinska-Nowak et al., 2018, Tanwattana and Toyoda, 2018, Tsai et al., 2020; Schueller et al., 2020; Teague et al., 2021; Altan, et al., 2022; Villagra et al., 2023). However, while serious games can contribute researchers with useful evidence into how people

conceive disasters, there is poor understanding into representation of catastrophes within popular culture (Gampell and Gaillard, 2016; Safran et al., 2024). Some authors related game characteristics for several disaster games to the disaster risk reduction framework (mitigation, preparedness and recovery), highlight the need for further research into how characteristics game (mechanics, dynamics, narrative and content), player skills, motivations and social interactions contribute to improve learning performance (Gampell and Gaillard, 2016; Safran et al., 2024). In addition, Weyrich et al. (2021) draw attention to

the lack of robust scientific evidence of the potential of serious DRM games. This lack calls for the development of more detailed studies to check and prove the effectiveness of serious games to educate DRM related activities in the area of disaster risk reduction (DRR) (Weyrich et al., 2021; Safran et al., 2024).

The remainder of this paper is organised as follows: an overview of serious games as tool for learning and change, especially those for disaster risk management (Section 2); a description of the qualitative methodological approach used (Section 3); a

presentation of qualitative content analysis of serious games and focus group with experts results (Section 4), and finally, a discussion of the results and main conclusions, including the limitations of this study and recommendation for futures researches (Section 5).

## 2. Theoretical framework

### 2.1 Serious games for learning and change

A large body of research supports the idea that active learning can improve learning performance more than traditional learning strategies, with gamification being one of the more representative examples (e.g. Tolks et al., 2024). Gamification refers to the use of game design elements in non-game contexts, in order to enhanced learning and certain behaviour (e.g.



Ramírez-Cogollor, 2014). Games awaken engage and motivation, provide social and civic skills and promote problem-solving capabilities (e.g. Liao et al., 2023; Safran et al., 2024). Today it is increasingly that educational platforms incorporate game elements (points, badges, difficult levels, leaderboards) so as to measure and encourage learning outcomes by adding scores and feedback (Hellín et al., 2023).

Serious games are those that have the purpose not only of entertaining, but also teaching, as well as conveying ideas, values, and influencing the thoughts and actions of players in real-life contexts (e.g. Frasca, 2007; Bylieva and Lobatyuk, 2019; Sáiz-Manzanares et al., 2021). The terms serious games and game-based learning are frequently used synonymously (Corti, 2006) although serious games have been created for the broader intentions of training and behaviour change in various fields, including business, healthcare, NGOs and education (Sawyer and Smith, 2008). Serious games are also referred to as "change games" (Bogost, 2007; Courbet et al., 2016) and "social impact games" (Cremers et al., 2014).

Serious games have experienced a rapid increase over the last decade, with extensive research supporting empirical evidence of cognitive benefits (Vogel et al., 2006; Bellotti et al., 2013), along with the identification of the impact on affective and motivational outcomes (Connolly et al., 2012; Wilson et al., 2009; Pineda-Martinez et al., 2023). Serious games allow users to visualize and explore phenomena that would otherwise be very difficult to experience, and see the consequences of their actions at different times (Wiek and Iwaniec, 2014). One of the great reasons for the effectiveness of learning through serious games is the immediate feedback they provide. These kinds of games allow learning through problem solving in an active way, where students focus solely on their learning (Medina, 2012). In addition, serious games favour personal autonomy, and social and cultural engagement (Magnuszewski et al., 2018).

A special case is that of serious games based on computer technologies, which have experienced a rapid increase in the last decade, increasingly replacing traditional games. Computer games take advantage of young people's interest in social networks and video or online games, and can cover diverse learning objectives, multiple fields and target various age groups (Mouaheb et al., 2012). Playing computer games is related to a variety of cognitive, affective, behavioural and motivational impacts and outcomes, the most frequent of these being knowledge acquisition and content comprehension (Connolly et al., 2012). Attending to their characteristics, there are a wide variety of genres and formats such as simulations, which simulate aspects of a real or fictional reality, and adventure, where the user solves challenges by interacting with people or the environment in a non-confrontational manner (Lamb et al., 2018; Heintz and Law, 2015; De Freitas, 2018; Heintz and Law, 2018).

**2.2 Serious games for disaster risk management**

Most disaster-related serious games involve social simulations and role-plays (Solinska-Nowak et al., 2018; Cremers et al., 2014). These types of games are intended for a high number of people, from different contexts, providing them with face-to-face discussion and negotiation about a given problem. Players have the opportunity to share different values and perspectives, engaging stakeholders with conflicting interests to cooperate towards a common goal (Akhtar et al., 2020).



Floods (e.g. Teague et al., 2021; Tsai et al., 2020; Gordon and Yiannakoulias, 2020), earthquakes (e.g. Safran et al., 2024; Feng et al., 2020; Whaley, 2019) and droughts (e.g. Podêbradská et al., 2020; Wang and Davies, 2015) dominate the subject of the different natural hazard games (Solinska-Nowak et al., 2018). This is in line with actual occurrence statistics, as, at least until 2015, floods have been the most common natural hazards globally over the last 20 years (43% of all events), followed by storms, with earthquakes in third place (8% of the total) (CRED, 2015). On the other hand, the three dominant

themes correspond to those causing the highest number of deaths (IFRC, 2020), and it is reasonable that they are the most represented in serious games.

Most serious games aim to strengthen the preparation capacity for natural risks (Solinska-Nowak et al., 2018). These games provide instructions through appropriate activities in relation to buildings, preparing emergency kits, stockpiling equipment and supplies and how to recognize the first signs of disasters (e.g. Tanwattana and Toyoda, 2018; Teague et al., 2021

Mossoux et al., 2016). In contrast, there are fewer games that focus on the post-disaster phase, which takes into account evacuation management (e.g. Feng et al., 2020) or how to save people (e.g. Whaley, 2019).

According to Solinska-Nowak et al. (2018), serious games achieve a broad range of public. Most serious games are focused mainly on adults, and to a lesser extent on younger people. This audience diversification constitutes a powerful tool for communication and education about DRM.

Serious games provide a satisfying learning and training experience of disaster management (e.g. Safran et al., 2024; Zhao et al., 2023). However, some limitations have been described that can potentially limit their effectiveness. Firstly, although serious games are destined to a wide audience, few examples consider cultural diversity, gender equality and learning from past events (Solinska-Nowak et al., 2018). This limitation is important because adequate risk management demands participatory strategies involving communities (Tanwattana and Toyoda, 2018). Instead, few studies have addressed the

development of diverse resiliency skills through serious games (Villagra et al., 2023; Teague et al., 2021; Neset et al., 2020), with the biggest research gap in serious games related to DRR is the lack of empirical evidence about their effectiveness, with a scarcity of quantitative and qualitative surveys (Solinska-Nowak et al., 2018; Safran et al., 2024).

## 3 Materials and methods

The approach to this study is qualitative, implies methodological triangulation and consists of primary research, supported by

secondary research. The methodological triangulation employed in this paper permits an issue to be studied from more than one standpoint, thus creating a stronger and more complete account. The research methods include a content analysis of selected serious games, a focus group with experts and a literature review (which enabled the research constructs to be determined).



### 3.1 Qualitative content analysis of serious games

In order to answer the first research sub-question ("How do serious games communicate and educate on issues related to natural hazards?"), a qualitative content analysis was carried out on serious games for DRM.

We have selected digital games from the wide range of existing examples available. Firstly, we conducted a web search using common search engines including Yahoo, Google, YouTube, Vimeo and the Apple iTunes store, using different combinations of the following keywords: serious games, positive communication, simulation, role play, DRM, DRR, crisis

management, emergency, disaster prevention and disaster mitigation. This search allowed us to find a total of 11 mobile apps, 6 online games and 20 board games with material available for download from the web (Table A1). Among the apps and online games we found 4 dedicated to volcanoes, 6 to earthquakes, and 7 related to floods.

Subsequently, in order to limit the scope of this study, we selected only non-commercial games with freely accessible content available in English or Spanish, with the necessary requirement of having a DRM focus in different situations, aimed

at young people (age 12 and upwards), with the disasters considered limited to those caused by human interactions with natural hazards (such as volcanic eruptions, earthquakes, floods, droughts, tsunamis, etc.). In this phase, a total of 6 games were selected: 3 mobile apps and 3 online games. Among the former, we found two flooding apps (Geostorm and Disaster Rescue Service) and one focusing on earthquakes (Earthquake Relief Rescue). Regarding the latter, two of the online games considered various natural hazards in a detailed way at different levels (Disaster Master and Stop Disasters), while Build a

Kit deals with emergencies due to natural hazards-related disaster in a more generic way.

A content analysis of the selected games was then carried out, which is a research tool used to analyse the presence of certain words, themes or concepts. Its use permits the quantifying and analysis of the presence, meaning and relationship between specific words, themes or concepts, and thus inferences to be made about the messages within the texts (Twining et al., 2017). Content analysis has several applications that make it very useful in answering the sub-questions posed. First, it is

able to identify the intentions or communicative tendencies in the games; in turn, it describes the attitudes or behaviours that result from those communications, revealing patterns in communication content. The dimensions analysed in this study were those proposed by (Ouariachi et al., 2017). These authors designed an instrument of analysis and evaluation of games about climate change with 51 criteria, classified into five dimensions: identification, narrative, contents, gameplay and educational aspects. These criteria are described further in Table B1.

### 3.2 Focus group with experts


In order to answer the second research sub-question, ("What are the educational and communicative elements or characteristics that serious games should have to improve DRM? an expert focus group was created. An ideal focus group is composed by 6-10 participants and is guided by a researcher who promotes participation. Focus groups are useful to supply information on participants' opinions and feelings about an issue and to asses the cause for their point of view (Twining et

al., 2017). Three characteristics allow achieving this: the open-ended question format, the closed environment of exchange



and discussion, and the facility for participants to share their opinions with others with the same interests and concerns. (Jayanthi and Nelson, 2001).

Firstly, a literature review allows determining the constructs of investigation. A combination of academic databases, including Web of Science, Scopus and Google Scholar was used. The web search focused on identifying available research
regarding the role of serious games in raising awareness of natural hazards and improving DRM.

The expert panel was very carefully chosen on the basis to knowledge or skill in the areas of either natural risk and education or videogames. We used a snowball sampling, asking the selected experts to recommend others who also matched our criteria. This study used individuals with previous knowledge of the topic being investigated, comprising an informed panel thus enabling the application of the title "experts" (Mullen, 2003). The required criteria to the participants was to have at
least of 5 years' professional or experiential of the topic of this research work. Twenty-one international experts (from Spain, Italy, Brazil and USA) took part in the study, with the main areas of expertise of the participants structured into videogames (8 participants) and natural risk and earth science education (13 participants) and the communication process was conducted online. The 62.5% of the videogame respondents were experts in video game design and the rest belonged to the areas of video game programming, development and production. As regards the surveys of experts in natural hazards, they present
different natural risk backgrounds such as climate change, floods, earthquakes, volcanoes and mass movement. By starting participants were informed through email about focus and the approach of this research including the subject, goal, focus group description, planning and ethical issues, confidentiality and anonymity.

The questionnaires were created with an open-ended question format and in a consensual way. The aim of this consensus was for the questions to allow the identification of indicators and criteria for the design of serious games on natural hazards
in order to improve DRM. For this reason, we endeavoured to include specific questions, eliminating those that were similar, avoiding questions that were too open-ended and focusing on those that allowed for a relevant response to the topic of study. Two online questionnaires were created using Google forms, one addressed to natural hazards experts and earth science educators, and one addressed to video games experts. Once all the responses had been collected, indicators were formulated based on their analysis using the MAXQDA program.

**4 Results**

**4.1 Qualitative content analysis of serious games**

Using the dimensions shown in Table B1, the results of the evaluation of each indicator are presented below.

**4.1.1   Identification**

The results of the identification dimension can be found in Table B2. We detected some differences between online games in
regard to mobile apps. The selected game apps for phones and tablets are created by individuals; in contrast, in the case of the online games, they are created by institutions such as the US Government and the United Nations for Disaster and Risk





Reduction. The language available in the games is English except in the case of Geostorm, which has a storyline based on a popular movie, and gives the option of a large number of languages, including Hungarian and Turkish. The communicative objective of the games analysed is to become familiar with natural hazards in general, to raise awareness of causes and
consequences, to promote changes in attitude and develop ideas for action and prevention.

### 4.1.2    Narrative

From the data examined, we observed different storylines, very different scenarios and a diversity of characters (Table B3). We found characters shared between two of the games, as in the case of Build a Kit and Disaster Master, meaning that students could therefore see it as a continuation of the story and connect more with the game, already know the characters.
The scenarios used are diverse, with some as familiar to students as a teenage girl's bedroom or the living room and bathroom of a family home (Build a Kit), and others as removed as an international space station (Geostorm). Both cases could contribute to student motivation. Recognition of the familiar setting can increase empathy with and thus awareness of the situation. However, the surprise of the space station as a more spectacular setting can lead to greater motivation.

The game development locations cover different points of the planet. In the case of Disaster Master, a game created by the
US government, the action takes place in different scenarios, all in US territory. Geostorm situates the player in different parts of the world such as Afghanistan, Dubai (United Arab Emirates) and Florida (US), according to the film on which it is based. Stop Disasters occurs in different parts of the world, depending on the natural catastrophe chosen, coinciding with the areas with the highest incidence of this natural hazard. This element has great educational value, as each game situates players in a real area of the planet they can relate to the natural catastrophe faced, which is fixed from the beginning of the
game. As all these places are real, the sensation of the real effects and consequences is easier to assimilate, therefore also facilitating awareness and promoting the acquisition of knowledge.

Present-future connection is addressed in most of the games, allowing players to be aware of the impact of their decisions and to experience this directly through the game. Only Earthquake Relief Rescue and Disaster Rescue Service focus their gameplay on a natural catastrophe that has already occurred and therefore the mission is limited to finding the injured, so
players know the consequences. The situation of Geostorm is similar, with the natural catastrophe being in process and the consequences already experienced in the game, but action can still be taken to stop the catastrophe and restore normality. As can be seen, being based on a fictional film, Geostorm the game represents that fantasy and is less realistic than other games such as Disaster Master.

Finally, considering the types of players, we can highlight the great diversity present in the games analysed. The main
character of Build a Kit is a girl in a wheelchair who faces the task of appropriate selection of utensils in an emergency situation, and Disaster Master presents the same character in a summer camp surrounded by her friends, each of a different origin. In the case of Geostorm, it also features both female and adult male characters in the role of superheroes.



### 4.1.3    Content

The results of the content dimension analysis are presented in Table B4. As for the terms used in the games evaluated, we found rather alarmist examples such as "emergency", "catastrophe" and "disaster". We did not find the term "natural risk", but rather "natural catastrophe" or "natural disaster". It is interesting to note the absence of misconceptions, although most of these games do not focus on explaining any complex concepts.

The link to social networks in some of them is merely informative, as in the case of Geostorm, which directs to the trailer of the film on which it is based. In the case of Build a Kit and Disaster Master, we find that the link provided to the government website leads to a space where we can find more information on natural hazards, as an extension of the knowledge provided by the game.

In general, the majority of the games (Build a Kit, Disaster Master and Stop Disaster) are presented in basic images with soft and cheerful colours. The game with the best image quality and effects is Geostorm, which, simulating the special effects of the film, contains more complex images, although their colours are darker than in the rest of the games, giving a more catastrophic feeling

### 4.1.4    Game play

The results of the game play dimension are shown in Table B5. The online games could be used both individual and collective way, while in the case of the mobile apps only individual play is possible. The objective of Build a Kit is to select utensils from different scenarios, therefore decisions can be made through teamwork. Disaster Master consists of reading a story in the format of a comic book and answering questions in order to check knowledge acquired individually or as a team. Finally, in Stop Disasters, based on which buildings to build or improve, where to build hospitals, or which barriers to build against the natural hazards, can also involve teamwork, thus making the game even more enjoyable. Therefore, these three online games offer further educational advantages over the mobile app examples.

The player trait most represented in the games evaluated is explorer, with the creative trait also being found in Stop Disasters, where there is a high level of interactivity over the course of the game. These two characteristics are related, as it is necessary for players to interact with the environment to explore what is happening around them.

In terms of game duration, there is a high degree of variability. The only example that could probably be completed in one hour is Build a Kit. Regarding the presence of levels rather than the whole game, the other games might also be played in one hour. Disaster Master could also be completed in one hour, depending on student age and level of comprehension.

Positive feedback abounds in the games and we only found a reward system in the online examples. None of the games offer the possibility of saving progress in the middle of a level, but if you complete a mission or level you can then continue in the next level at a later time, with the exception of Build a Kit.



### 4.1.5 Learning implications

The results of the analysis of the educational dimension (Table B6) show the great potential of the games evaluated, which
cover a broad spectrum of the cognitive levels of Bloom's Taxonomy (reviewed by Anderson et al., 2001). The most
complete is Stop Disasters, with Earthquake Relief Rescue and Disaster Rescue Service being the two games that cover only
the lowest cognitive levels.

Likewise, Disaster Master and Stop Disasters permit working with the key competences for lifelong learning according to
the European education curriculum (European Commission, 2019). All the games evaluated enable the obtaining of
Citizenship Competence and Digital Competence. Build a Kit and Disaster Master enable the achievement of the Personal,
Social and Learning to Learn Competence. In contrast, it is only possible to relate the Literacy competence to Disaster
Master. Earthquake Relief Rescue and Disaster Rescue are the only games that do not work on the Mathematical, Science,
Technology and Engineering (STEM) competence. The Multiculturalism and Cultural Awareness and Expression
Competences are addressed by the online games.

**4.2 Expert focus group**

**4.2.1 Videogames experts answers**

Most of the videogame experts (63%) agree that the catastrophic/dramatic and adventure theme, teaching protocols for
action and a possible escape dynamic are key elements in the design of a video game to raise awareness of natural hazards.
In line with this catastrophic theme, some experts (33%) believe that the inclusion of the danger to human life element could
enhance empathy on the part of players. Some experts (38%) add that it is also necessary for the game to be generally fun for
children to want to play, and that it is always the message of the game that is fundamental to the educational intention (Table
1).

**Table 1.** Results of responses from video game experts.

| | no. of votes | Importance (%) |
|---|---|---|
| **Elements** | | |
| Catastrophe | 5 | 63 |
| Protocols | 5 | 63 |
| Adventure | 5 | 63 |
| People/lives | 3 | 38 |
| Narrative/message | 4 | 50 |
| Escape | 5 | 63 |
| Fun | 3 | 38 |
| **Character** | | |



| Socialiser | 5 | 63 |
|---|---|---|
| Explorers | 3 | 38 |
| **Reward systems** | | |
| Secondary | 3 | 38 |
| Recommended/satisfactory | 2 | 29 |
| Disconnect | 2 | 29 |
| Fun | 1 | 13 |
| Always | 2 | 29 |
| Engaging | 3 | 38 |
| **Levels** | | |
| Distract | 1 | 13 |
| Engage | 3 | 38 |
| Progress/motivation | 4 | 44 |
| **Duration** | | |
| Hours | 3 | 35 |
| Short | 1 | 13 |
| Minutes | 4 | 46 |
| Narrative | 3 | 38 |
| Engage | 3 | 43 |
| Variable | 2 | 29 |

The majority of the videogame experts (63%) agree that the most interesting type of character would be the socialiser in a
cooperative game dynamic. They remark that for raising awareness of natural hazards, interacting and collaborating with
other players who have been affected by the disasters could help to generate empathy. Similarly, some of them (33%) state
that the profile of explorers is very interesting for this topic, since, being a natural phenomenon, it is also convenient for
interaction with the environment (Table 1).

The videogame experts recommend including reward systems because they increase engagement and fun (Table 1).
However, some (38%) remark that the reward system could be considered as a secondary element, claiming that by including
it players may focus on the rewards and disconnect from the main objective, which is to raise awareness and increase
knowledge about natural hazards. In the same way, according to some experts (13%), the abuse of the employment of levels
and progression bars could distract from the main objective.

In order to increase knowledge about natural hazards, it is also important that feedback systems be included to make players
feel connected to the game. Some experts (46%) agree that positive feedback is the most effective for motivation, but it
would be interesting to include both positive and negative feedback so that players can see their actions have both good and
bad repercussions (Fig. 1). In this regard, the experts point out that the age of the students must be taken into account; if they





are too young, it is more convenient to include positive feedback. However, as they advance in development and enter adulthood, it is advisable to include both. They also emphasise that in order to promote the acquisition of knowledge,

feedback should always be constructive.

Regarding the duration of play, it should be a relatively short game, between 15 minutes up to several hours. This duration is recommended for both the full game and for the individual levels. The game can be longer, as long as it has concrete levels or scenarios that can be completed in a short period of time. Some experts (38%) emphasise that the most important element is the narrative of the game, which must engage players and that, where this is achieved, the length of the game can even be

variable (Table 1).

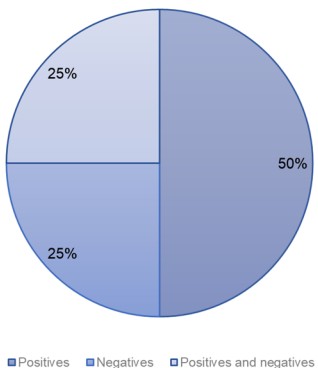

**Figure 1.** Responses from video game experts on the game feedback system. Each sector represents the relevance rating.

All of the experts agree that the game should have aspects based on curiosity, since curiosity leads to research, and research provides knowledge that allows us to provide solutions (Fig. 2). However, these aspects should be optional or very well

integrated so that they are not forced or heavy. There are various responses to the question of randomness, with the experts commenting that chance aspects should not be included, as this diverts attention from what is important and leads players to experience the wrong stimuli. However, an abuse of chance can be frustrating for players, who feel they have no control over the game and may cause them to drop out, so this element should be balanced and not excessive.



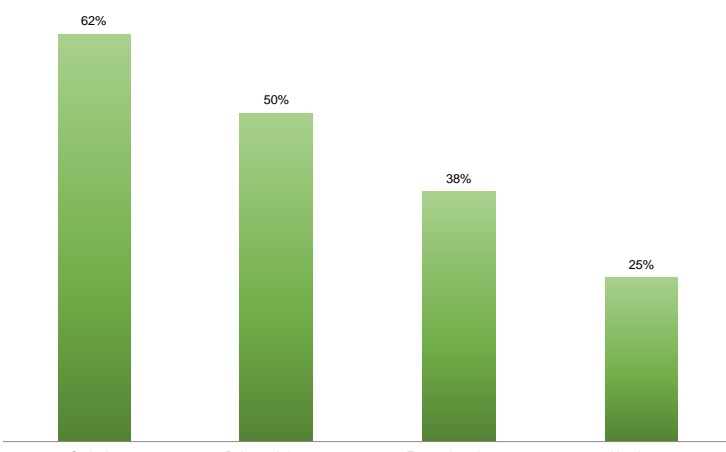

**Figure 2.** Responses from videogame experts on curiosity and chance. Each bar represents the relevance rating.

Finally, the experts recommend there should be interaction in the game (Fig. 3), as it generally engages players and makes it more fun. However, the code most represented in this question is narrative, since, in a serious game the objective is transferring knowledge, for which narrative is essential.

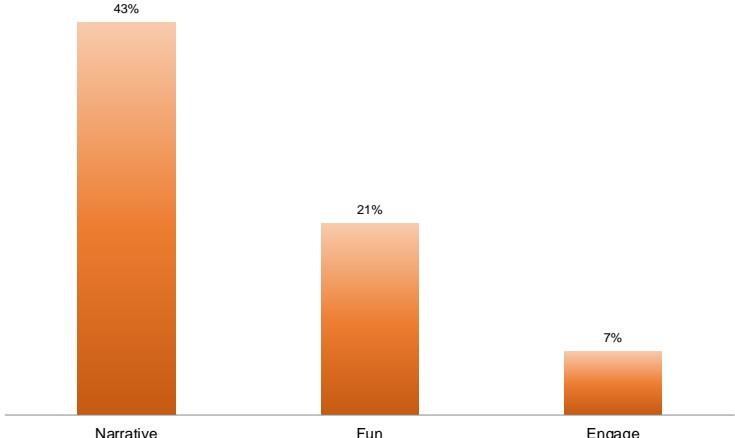

**Figure 3.** Responses from videogame experts on the level of interaction. Each bar represents the relevance rating.

### 4.2.2 Natural risk expert answers

These experts criticise the media for being too alarmist, leading to uncertainty and focusing only on high-impact disasters once they have happened, which fails to enhance collective and individual prevention. In addition to this, they rarely contact university or technical experts, so they resort to misleading clichés. Therefore, the information they transmit does not contribute to raising public awareness or to the acquisition of fundamental knowledge about natural hazards (Table 2).





**Table 2.** Results of responses from natural hazards experts.

|  | no. of votes | Importance (%) |
|---|---|---|
| **Comunication** | | |
| Alarmist | 6 | 46 |
| Deficient | 8 | 62 |
| **Inequalities** | | |
| Determining inequalities | 13 | 100 |
| Dependent vulnerability | 2 | 15 |
| **Literacy/awareness** | | |
| Education | 11 | 85 |
| Age-dependent | 4 | 31 |
| **Multiculturalism/gender** | | |
| multiculturalis | 10 | 77 |
| gender | 10 | 77 |
| Not important | 3 | 23 |
| **Character** | | |
| Normal person | 6 | 46 |
| Eligible avatar | 1 | 8 |
| Not important | 1 | 8 |
| **Sources of information** | | |
| Academic | 11 | 85 |
| Official institutions | 2 | 15 |
| Historical | 1 | 8 |
| Scientific communicators and journalists | 2 | 15 |

The experts also agree that social and structural inequalities significantly condition the vulnerability of a territory or a society, since livelihoods, housing, etc. are often more vulnerable to natural hazards as well as access to training and information also being unequal. Therefore, a balanced society will be much more resilient as a whole than an unequal society

(Table 2). The experts suggest that video games could help to better understand natural hazards and raise awareness among players, i.e. they could be good training in prevention and vulnerabilities to natural hazards, providing information on self-protection, planning and emergency management in a playful and enjoyable way (Table 2). Aspects of multiculturalism and gender should be included in the game, which must consider interracial, intercultural, disability and gender factors, as these are fundamental for any society. All cultures and genders should be included in video games given that the whole of society,

without exception, can be affected (Table 2). For this reason, the main character of the game on natural hazards should be an




ordinary, responsible, coherent and supportive person, who has fears and faces them by learning, and who can also fail. In other words, a character with whom players can see themselves represented (Table 2).

Figure 4 shows the results obtained from the experts' answer in regard to the game narrative and context. Most of the experts opt for simple, non-technical narratives in order for players to be familiar with the language and feel that it is a real situation.

They mention the figure of a saviour character in the face of natural hazards, making them aware of the resulting environmental and social problems, as well as the figure of a researcher who goes to the area of the event to investigate and advise on the situation. Less represented is the planning of a new human settlement, taking into account the location of infrastructures, buildings and the type of construction, providing the player with valuable knowledge. The last code present among the expert responses is the representation of past episodes related to natural hazards-related disasters that have

occurred through time. In this way, they convey that natural hazards are things that have always existed and will continue to exist, reinforcing the idea they are real and have happened at different times in human history.

The experts propose different sources of information to be used in natural hazard games such as historical sources and those from official institutions. However, the most important example is the academic source, in order not to introduce erroneous data or information (Table 2).

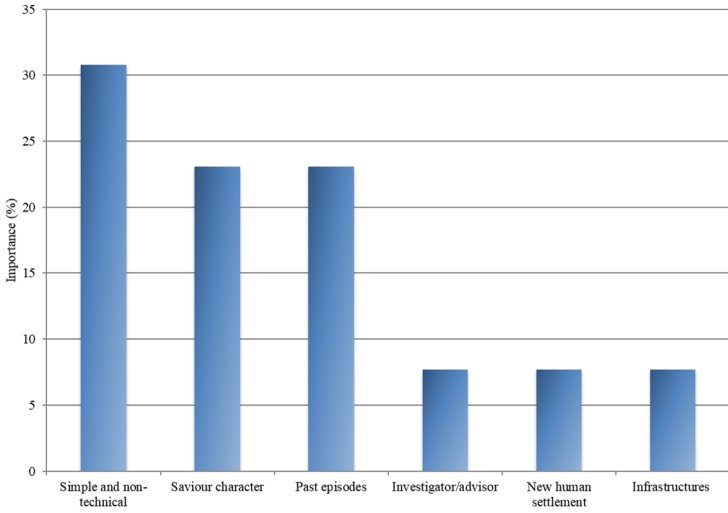


**Figure 4.** Responses from natural risk experts on narrative and context. Each bar represents the relevance rating.

The tone of the message that should be used in video games about natural hazards presents some controversy (Fig. 5). The experts opine that an informative tone is important for transferring the information combined with an emotional tone to have a greater impact on the user, helping to raise awareness. All of the experts reject the alarmist tone with the exception of two,

who propose mixing the informative tone with the alarmist tone to prevent the former from being boring and causing indifference. Experts who suggest a purely informative tone also propose a clear and concise message based on science, so that players know what can really happen and how to act in an objective manner.



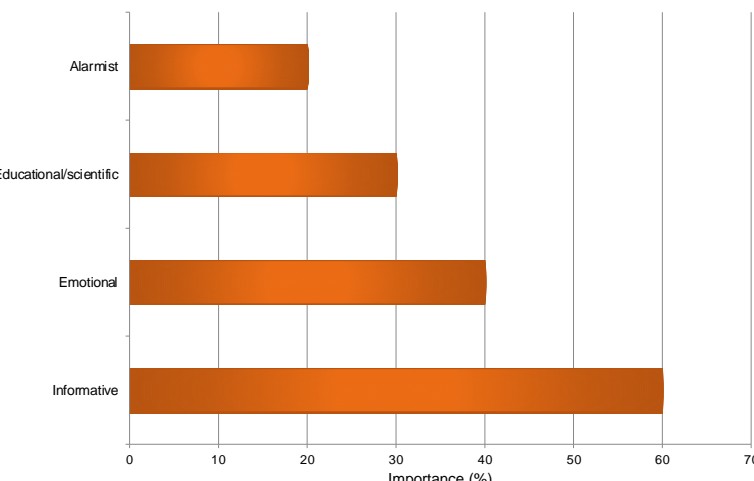

**Figure 5.** Responses from natural risk experts on message tone. Each bar represents the relevance rating.

## 4.3 Integration finding

In this section, the findings of the methodological triangulation were compared – literature review, qualitative content analysis and expert focus group – and further summarised:

### 4.3.1 Characters

The main characters of the game should be socialisers or explorers with characteristics of an ordinary person who reflects their fears and may fail, who also presents a saviour attitude. Multicultural and gender aspects should be included in the characters, along with the consideration of interracial and disability aspects. Games on natural hazards should take into account gender and cultural differences in order to reflect today's society.

The only game with a socialising character in a cooperative dynamic is Disaster Master. The three mobile apps, Geostorm, Disaster Rescue Service and Earthquake Relief Rescue all feature more exploratory characters. In this sense, Disaster Rescue Service and Earthquake Relief Rescue do present a cooperative dynamic, but there is no opportunity for interaction between players.

### 4.3.2 Information and message tone

Information should be presented in a non-alarmist and non-catastrophist way. The sources of information used for the development of the games should be mainly academic. The tone of the game message should be clearly informative, clear and concise. It should also have an emotional tone to connect with players.

The mobile apps present a more catastrophist and alarmist tone, with people's lives endangered and the adventure factor increased but without working on the narrative. However, Disaster Master also plays with this point of catastrophism and fun, and at the same time has a wonderful narrative that aims to transmit knowledge and teach protocols for action. Stop





Disaster has great message texts explaining in a very clear, concise and simple way the usefulness of each material to

prevent the impact of natural hazards and how they should be used. Build a Kit bases its game on teaching protocols of action against a natural hazards-related disaster.

### 4.3.3 Narrative, dynamics, reward systems and feedback

The narrative of the game should be simple and non-technical, and could represent past episodes. The videogame experts recommend a narrative based on a catastrophic/dramatic and adventure theme. The dynamics of the game should be

cooperative. Reward systems and levels or game progression bars should be included secondarily. Feedback should be included taking into account user age, and positive feedback is especially important. The duration of the games should be between 15 minutes and several hours. The game always should be fun.

Build a Kit, Geostorm, Disaster Rescue Service and Earthquake Relief Rescue present only one reward system, which consists in completing the level or scenario where you are, choosing the right tools to make an emergency backpack, opening

the office door to escape, or arriving in time to rescue an injured person. The other two games have game progression indicators and award points to the player when they choose the correct answer or construct a building in a suitable location, but always secondarily. All of the games have feedback systems for the players, and their lengths are within the recommendation of experts.

## 5 Discussion and Conclusion

### 5.1 Theoretical and practical implications

In response to the need for engaging and motivational approach to education and communication, videogames have been recognised as one of most useful strategies in teaching DRR (e.g. Hawthorn, 2021). However, the impacts of video games on players in order to improve DRM in citizens remain relatively unstudied (e.g. Safran et al., 2024). This study therefore presents new insights revealing how serious games could communicate and educate DRM more effectively.

Serious games employed to communicate and educate about natural hazards present narratives are highly varied. In most of these games a connection is made between the present and the future, allowing players to be aware of the impact of their decisions and to experience them directly through the game. The central character of the game is usually an explorer. Few games consider the multicultural aspect, with only Build a Kit and Disaster Master presenting characters with different backgrounds and individuals with functional diversities, which can be connected to the previous findings of Solinska-Nowak

et al. (2018) in relation to multiculturalism in natural hazard games. The inclusion of people with functional diversities, and even more of young adolescents as the target group of the game, and of young people from different backgrounds and cultures, no longer only as characters in the game, but as a group of friends who together provide a solution to a situation of risk is important. In this way, awareness-raising is much richer, as it encompasses aspects of knowledge of natural risks, and also of tolerance, inclusion and social empathy. Geostorm present characters, both women and adult men, in the role of





superhero, breaking the gender gap that we can easily find in many games where the hero figure is attributed to the male gender. Therefore, games can offer a learning of values of great social importance, which is undoubtedly of considerable educational interest. Taking into account the learning performance, the results of the educational dimension show the great potential of games, which cover a broad spectrum of the cognitive levels of Bloom's Taxonomy (reviewed by Anderson et al., 2001). The results obtained are intended to provide valuable guidance to teachers in selecting games to implement in the

classroom.

In order to know how to communicate and educate more effectively in regard to natural hazards through serious games, this work determines the desirable characteristics that games should have through expert interviews. These desirable characteristics are: exploratory characters in a cooperative dynamic, with simple and non-technical narratives, based on academic sources, multiculturalism, consideration of diversity and gender, fun, short games, constructive feedback system

and rewards have secondary presence. These findings can be considered by videogames designers in order to create new natural risk games that improve DRM.

The expert opinions are in line with the results of other studies on how to promote social resilience (Tanwattana and Toyoda, 2018; Kwok et al., 2016). In this sense, resilience could be encouraged with a cooperative dynamic, where democratic and collaborative decision-making and problem solving occur and community beliefs and values are shared, thus promoting

collective efficacy. The use of simple and non-technical narratives based on academic information could promote natural risk knowledge and hazard consequence and therefore community preparedness for natural risk. The multiculturalism, diversity and gender awareness that should be considered in the games could favour community inclusiveness in DMR and encourage a sense of community and attachment.

Regarding the game characters desirable in DRR games, Safran et al. (2024) reveal that the performance of players is related

with the video game narrative and highlight the importance of character characteristics, so, high-powered avatars lead to a greater increase in attempts to adopt health-promoting behaviour. Klimmt et al. (2009) assert that players undergo significant changes in self-perceptions to align themselves with certain characteristics of such characters. Therefore, the identification with the character that players undergo can increase perceived self-efficacy concerning the acquisition of health-promoting behaviour (Peng, 2008).

Considering elements desirable in DRR games such as feedback and rewards, the feedback is more efficient when it is based in a direct and specific information that achieve objectives and is presented close to that of the item being evaluated, and this feedback should be both positive and empowering (Prensky, 2001). Rewards systems are recommended because they enhance motivation and entertainment; however, too much of these could distract from the main objective of the game (Chou, 2015). In addition, an engaging game should be entertaining, in this way, players are more likely to play the game

multiple times, which will keep the issue active in their minds (Ouariachi et al., 2019).

Controversy of criteria between experts in relation to the tone of the message has been found. The natural risk experts recommend a non-alarmist and non-catastrophist tone, while the video game experts agree on a catastrophic/dramatic tone. In this regard, self-efficacy may be diminished by the panic and stress provoked by perception of both the gravity of a hazard





and one's own susceptibility to it, which is important for motivating risk mitigating action (O´Neill, 2004). The natural risks
frequently provoke negative emotions, including denial, fatalism, which are counter to the problem orientation necessary in
trigger risk mitigating actions (Safran et al., 2024). However, Zhao et al. (2023) show that these negative emotions are
necessary because they have a greater impact on decision-making than the positive emotional state. Therefore, well-designed
video games could balance threats by offering ways to overcome them, incorporating mediated disaster-related problem-
solving experiences (Safran et al., 2024).

The recommended features set out in this work were tested in the selected online games and mobile apps. Only the online
games comply with the fundamental narrative highlighted by the experts in order to fulfil the educational function of the
game. The apps focus their game dynamics more on the interaction, both of the player with the game controls and of the
character with the environment. This interaction could enhance the fun aspect of the game and therefore increase desire to
play. Regardless, the educational aspects of online games is much greater, both in the explicit knowledge of the messages of
the game, as well as in its dynamics and progress. In addition, only three online examples (Build a Kit, Disaster Master and
Stop Disasters) work on issues of multiculturalism, diversity and gender, as well as geographically locating the areas where
natural hazards are most likely to occur.

**5.2 Limitations and recommendation for futures studies**

Qualitative analyses focus on interpreting meaning and the meaning-making process, thus inherently possessing a subjective
nature. Consequently, a methodical approach of triangulation was used to minimize this limitation. In addition, the authors
analysed the data through several rounds, seeking consensus among the established codes.

This study is exploratory in nature, thus we encourage researchers to delve deeper into how videogames can enhance DRM.
Further research could enhance our understanding of how specific narrative elements, mechanics, and characters in disaster-
related video games improve DRM. In this sense, subsequent studies could concentrate on validating the effectiveness of the
proposed features in enhancing DRM among citizens.

**Appendix A. Games on natural hazards.**

**Table A1**. Games on natural hazards found in the literature review.

| Game | Supplier / website |
|---|---|
| *Phone and table apps* | |
| Sai Fah - The Flood Fighter | Opendream Co., Ltd. |
| Earthquake Safety Tips - How To Protect Yourself | BABYBUS CO., LIMITED |





| Seismic Safety - earthquake protection (available only in Chinese) | Zhi Yong Information |
|---|---|
| Earthquake Relief & Rescue Simulator | Atif Mumtaz |
| The Earth by Tinybop (payment required for download) | Tinybop Inc. |
| How It Works? (payment required for download) | Learny Land S.L. |
| Pompeii Run Volcano Escape | Big Goose Egg, LLC |
| Volcano Defense | Adknown Inc. |
| Geostorm | Talespin, LLC |
| Emergency hero flood rescue | Adeel Ahmad |
| Lifeboat Rescue - Save Life | Atif Mumtaz |
| Disaster Rescue Service | Syd Umer Aftab |
| *Online games* | |
| Build a kit | https://www.ready.gov/kids/games/data/bak-english/index.html |
| Disaster Master | https://www.ready.gov/kids/games/data/dm-english/ |
| Earth Girl. The Natural Disaster Master (difficulty downloading game content required to play the game on the PC) | https://earthgirl2.com/level-and-character-art/ |
| Earthquake Response (difficulty downloading game content required to play the game on the PC) | http://www.enabledgames.com.au/stc/ |
| Extreme Event Game (min. 6 players) | https://learn.labx.org/ee-download |
| Stop Disasters | https://www.stopdisastersgame.org/ |
| Act to adapt | https://www.climatecentre.org/games/2541/act-to-adapt/ |
| Before the storm | http://petlab.parsons.edu/redCrossSite/rulesBTS.html#d ownloadsBTS |
| Buzz about Dengue | https://www.climatecentre.org/games/2530/buzz-about-dengue/ |



| | |
|---|---|
| Cultural Memory Game: Earthquake and Flood version | https://culturalmemory.socialsimulations.org/ |
| Board games with downloadable materials available | |
| Act to adapt | https://www.climatecentre.org/games/2541/act-to-adapt/ |
| Before the storm | http://petlab.parsons.edu/redCrossSite/rulesBTS.html#downloadsBTS |
| Buzz about Dengue | https://www.climatecentre.org/games/2530/buzz-about-dengue/ |
| Cultural Memory Game: Earthquake and Flood version | https://culturalmemory.socialsimulations.org/ |
| Decisions for the decade | https://www.climatecentre.org/games/2520/decisions-for-the-decade/ |
| Dissolving Disasters (especially because it involves communication and how to handle the moment) | https://www.climatecentre.org/games/2516/dissolving-disasters/ |
| Evacuation Board Game | http://www.floodsite.net/juniorfloodsite/html/en/student/thingstodo/games/boardgame.html |
| Evacuation Challenge Game | https://evacuationchallenge.socialsimulations.org/ |
| Evacuation Role Play Game | http://www.floodsite.net/juniorfloodsite/html/en/student/thingstodo/games/roleplayinggame.html |
| Flood Resilience Games | https://floodresilience.socialsimulations.org/ |
| Game of Floods | https://www.marincounty.org/depts/cd/divisions/planning/csmart-sea-level-rise/game-of-floods |
| Gender and Climate Game | https://www.climatecentre.org/games/2510/gender-and-climate-game/ |
| Gifts of Culture: Diversity in the context of flood resilience | https://giftsofculture.socialsimulations.org/en/ |
| Lords of the Valley: Experience and explore sustainable practices in complex environments | https://lordsofthevalley.socialsimulations.org/ |
| Paying for Predictions | https://www.climatecentre.org/games/2501/paying-for-predictions/ |
| Ready! | https://www.climatecentre.org/games/2497/ready/ |



| Riskland | https://www.unisdr.org/2004/campaign/pa-camp04-riskland-eng.htm |
| Save Natalie! The preparedness game | http://helid.digicollection.org/en/d/Jdnd24/5.html |
| Upstream Downstream | https://preparecenter.org/resource/the-upstream-downstream-game-2/ |
| Weather or Not | https://petlab.parsons.edu/redCrossSite/rulesWON.html |

**Appendix B. Qualitative content analysis.**

**Table B1.** Dimensions and indicators of qualitative content analysis.

| Dimension | Indicators |
|---|---|
| Identification | – Name of game<br>– URL/App<br>– Type of creator (author and type of institution): *name and type of institution involved in the game's creation, and placement of the URL within an independent website or in a section of the producer/author's webpage or another webpage.*<br>– Language<br>– Communicative Objective: *communicative intentions and goals.*<br>– Brief Description: *summary according to genre, objectives, and story.* |
| Narrative | – Narrative Relevance: *importance or irrelevance of narrative elements.*<br>– Global Story: *description of the game's narrative as a whole, based on the logical or causal succession of events over a specific period.*<br>– Character Representation: *role and characteristics of the character.*<br>– Environment Representation: *the setting in which the character operates.*<br>– Dimension / Space / Scale: *general context and scale of the scenarios.*<br>– Dimension / Time: *period covered by the story.* |
| Contents | – Terminology used to describe natural hazards.<br>– Presence of false concepts or misconceptions about natural hazards.<br>– Explicit use of scientific concepts.<br>– Convergence with other media and social networks: *links to social media platforms.*<br>– Explicit use of information sources: *citing sources and origin of data.*<br>– Message framework: *topics, causes/consequences, and tone of the message.*<br>– Images. |
| Gameplay | – Number of players and usage (individual or collective)<br>– Player type: *profile tailored to their interests.*<br>– Level of interactivity: *degree of user intervention, modification, and choice over the content.*<br>– Duration of game<br>– Game mission: *essential actions to win the game.*<br>– Feedback system: *comments through text, audio, or audiovisual means received by the player for certain actions.*<br>– Reward system<br>– Availability of instructions / possibility of saving game (yes, no) |
| Didactics | – Competencies: *Knowledge and attitudes attained by the student.*<br>– Skills: *Mental operations achieved by the student.* |





|  |  |
| --- | --- |
|  | – Conditions for problem-solving: *Type of reasoning employed to solve problems.*<br>– Need for prior knowledge<br>– Learning curve: *Level of difficulty in learning.*<br>– Possibility of group work: *Refers to the ability to form a group of students around the computers.*<br>– Accessibility: *Availability of the game for students with functional diversity.*<br>– Interdisciplinarity: *Combination of two or more academic disciplines.*<br>– Possibility of teacher evaluation: *The teacher can access the history of actions, intervention records, etc.* |

**Table B2.** Results of identification dimension.

| Name of game | URL/App | Type of creator | Language | Communicative objective | Brief description |
| --- | --- | --- | --- | --- | --- |
| Build a Kit | https://www.ready.gov/kids/games/data/bak-english/index.html | Official Website of the United States Government | English | Developing familiarity with subject matter and acquiring knowledge about how to react to disasters. | Various questions to answer that promote reflection on how to act in those circumstances. |
| Disaster Master | https://www.ready.gov/kids/games/data/dm-english/ | Official Website of the United States Government | English | Developing familiarity with subject matter and acquiring knowledge about how to react to disasters. | Various questions included in a narrative to answer in order to verify comprehension of story and promote reflection on how to act in face of natural hazards. |
| Stop Disasters | https://www.stopdisastersgame.org/stop_disasters/ | Website of the United Nations Office for Disaster Risk Reduction (UNDRR) | English | Developing knowledge of prevention to decrease probability of disaster. | Simulation in which player creates territory with lower probability of experiencing natural hazards, analysing buildings within it. |
| Earthquake Relief Rescue + | App | Atif Mumtaz | English | Developing knowledge related to topic. | Simulation in which victims of an earthquake are searched for and rescued in a city that has been devastated, with assistance of a dog. |
| Disaster Rescue Service | App | Syd Umer Aftab | English | Developing knowledge related to topic. | Simulation in which victims of a major flood are searched for and rescued using a boat or a helicopter. |
| Geostorm | App | Talespin | English, French, German, Polish, Portuguese, Russian, Italian, Spanish, Turkish, Chinese, Hungarian | Familiarisation with mode of operation. | Player surpasses various levels to escape from a building destroyed by natural hazards. |



**Table B3.** Results of narrative dimension.

| Name of game | Build a Kit | Disaster Master | Stop Disasters | Earthquake Relief Rescue + | Disaster Rescue Service | Geostorm |
|---|---|---|---|---|---|---|
| **Relevance of narrative** | Medium | High | Medium | Low | Low | Medium |
| **Global History** | The player joins Gayle and her friends in aiding them to gather supplies in preparation for an emergency. The player progresses through various levels in which they must select the appropriate tools for themselves and their family to survive natural hazards. | There are eight different levels, so if you score enough points to complete a level, at the end of it, you receive a password that allows you to move on to the next level. Each level deals with a different natural hazards. The overarching framework is the narrative of a comic-style story in which you have to answer various questions about natural hazards to earn points. | At the beginning of the game, you must choose a natural hazards: tsunami, hurricane, fire, earthquake, or flood; and thereafter build upon an already established community to provide defense and enhance structures for the inevitable natural hazards that is to come. | A major earthquake has struck the city, roads and buildings are destroyed, and there are various injured individuals in need of rescue. With the assistance of a trained dog, the task is to locate these injured individuals. | The player acts as a rescue services specialist, saving lives and assisting individuals affected by floods in a large city by transporting them to a safe area. | Following the sabotage of a network of weather satellites that protect the Earth, a series of meteorological catastrophes are destroying several cities worldwide. The game involves gathering essential data to prevent a geostorm and escape from the affected areas. |
| **Representation of character** | Gayle, a girl in a wheelchair, and her friends are a group of teenagers from diverse ethnic backgrounds. | Once again, it is Gayle and her friends: Raina, Sonny, Misti, Ray; in various scenarios where they encounter numerous natural hazards. | There is no character | A man and his dog, but in a very impersonal manner, without mentioning any names. | Various impersonal characters: helicopter or boat pilot, doctor, and rescuer. | A SataCorp office worker in Dubai, an astronaut stationed at the ISS IV, a communications officer in the special forces, and an atmospheric weather analyst. |
| **Representation of environment** | Different settings: Gayle's room, a family living room, a family bathroom, and a convenience | Each level unfolds within a distinct environment, contingent upon the phenomenon. The wildfire occurs at a summer camp in | Populations are depicted in the following locations according to the natural hazards: for | A city devastated by an earthquake. | Simulation of Los Cantos City | SataCorp offices in Dubai, International Space Station IV, a remote village in |



| | | | | | |
|---|---|---|---|---|---|
| store. | Colorado; the tornado strikes at a high school in Iowa; the hurricane ravages Louisiana; the house fire erupts in Connecticut; the winter storm/freezing occurs in Iowa; the tsunami hits Hawaii; the earthquake shakes California; the lightning storm occurs in a park in Iowa; and the heatwave does not specify a location. | the tsunami, one in Southeast Asia; for the earthquake, one in the eastern Mediterranean; for the flood, one in Central-Eastern Europe; for the wildfire, one in central Australia; and for the hurricane, one in the Caribbean. The populations are represented in a somewhat unrealistic manner. | | | Afghanistan, and a warehouse complex in Orlando, Florida. |
| **Dimension/ Space/ Scale** | Fictitious / local | These are real locations where events are simulated. | Real sites, fictional populations. | Fictitious / local | Fictitious / local | Fictitious / local |
| **Dimension/ Time** | Present | Present | Present | Present | Present | Present |

**Table B4.** Results of contents dimension.

| Name of game | Build a Kit | Disaster Master | Stop Disasters | Earthquake Relief Rescue + | Disaster Rescue Service | Geostorm |
|---|---|---|---|---|---|---|
| **Terminology employed to describe natural hazards.** | Emergency, disaster | Disaster, wildfire, tornado, hurricane/blackout, home fire, Winter storm/extreme cold, tsunami/earthquake, thunderstorm/lightning | Disaster, tsunami, hurricane, wildfire, earthquake, flood | Earthquake | Disaster, flood | Thunderstorm, alert, emergency |
| **Presence of misconceptions or errors regarding** | No | No | No | No | No | No |



| **natural hazards.** | | | | | | |
|---|---|---|---|---|---|---|
| **Explicit use of scientific concepts.** | No | Wildfire, tornado, hurricane/blackout, winter storm/extreme cold, tsunami/earthquake, thunderstorm/lightning | Tsunami, hurricane, wildfire, earthquake, flood | Earthquake | Flood | Thunderstorm |
| **Convergence with other media and social networks.** | Government website: https://www.ready.gov/kids | Government website: https://www.ready.gov/kids/disaster-facts | No | No | Facebook, Youtube | Youtube |
| **Explicit utilization of information sources.** | No | No | No | No | No | The film "Geostorm" |
| **Message framework** | *Themes:* Preparedness for emergency or natural hazards. *Consequences:* Not knowing how to react can lead to disastrous consequences. | *Themes:* Natural hazards can occur in any environment and situation. *Consequences:* Knowing certain actions such as prevention or reaction to natural hazards is necessary to avoid a catastrophe. | *Themes:* Defences and improvement of buildings to prepare for the inevitable disaster. *Consequences:* These actions can save lives. | *Themes:* An earthquake has shaken the city, leaving roads and buildings in ruins. *Consequences:* There are people trapped who will die if they are not rescued. | *Themes:* Excessive rainfall has flooded an entire city. *Consequences:* There are many people who need to be rescued at various points in the city via helicopter or boat. | *Themes:* A network of weather satellites protecting the Earth has been sabotaged. *Consequences:* Cities worldwide are being destroyed by meteorological catastrophes. Essential data must be collected to prevent a geostorm. |
| **Images** | Static drawings of a teenage girl's bedroom, a family living room, a family bathroom, and a variety store. | Drawings of various scenarios in a comic style. | Basic pixelated drawings of villages. | Scenario of a destroyed city. | Scenario of a flooded city. | Different scenarios previously specified in other sections with high-quality graphics. |



**Table B5.** Results of gameplay dimension.

| Name of game | Build a Kit | Disaster Master | Stop Disasters | Earthquake Relief Rescue + | Disaster Rescue Service | Geostorm |
|---|---|---|---|---|---|---|
| **Number of players and usage (individual or collective).** | 1, for individual or collective use | 1, for individual or collective use | 1, for individual or collective use | 1, for individual use | 1, for individual use | 1, for individual use |
| **Type of player.** | Explorer | Explorer | Explorer / creator | Explorer | Explorer | Explorer |
| **Level of interactivity.** | High | Medium | High | High | High | High |
| **Game duration.** | Depending on prior knowledge, from 15 minutes to 1 hour. | From 45 minutes to 1.5 hours or more. | One session, one hour; the entire game, days. | More than an hour. Days. | More than one hour. Days. | More than one hour. Days. |
| **Game mission.** | The aim of the game is to select the correct tools to survive a disaster; if you do not select them or choose the wrong ones, you will not progress to the next scenario. | The objective is to verify that the player understands how to react to disasters or prevent risks from the comic by asking questions about things that have been said in the story. If you answer too many questions incorrectly, you lose the level and have to start over. At the end of a level, you receive a password that allows you to play the next level. | The mission is to invest the starting money that the player has in improving the resilience of buildings, creating buildings to protect the entire population, or building defences. The game indicates the probability of the catastrophe occurring, which increases over time until it happens, and you check if you have saved lives with your modifications or not. | The aim of the game is to guide the rescue dog to the trapped person using the mobile phone / tablet controls. | The objective of the game is to use different vehicles to find people who need to be rescued. | The mission of the game is to guide the character through a scenario affected by a natural disaster, finding the exit from different rooms and searching for the documentation necessary to restore Earth's protection. |
| **Feedback system.** | Positive | Positive and negative | Negative | Positive | Positive | Positive |
| **Reward system.** | Yes, next level. | Yes, next level and game points. | Yes, congratulations on saving lives and game points. | No | No | No |
| **Availability of instructions/possibility of saving game (yes, no)** | Yes/No | Yes/No | Yes/No | Yes/No | Yes/No | Yes/Yes |



**Table B6.** Results of educational dimension.

| Name of game | Build a Kit | Disaster Master | Stop Disasters | Earthquake Relief Rescue + | Disaster Rescue Service | Geostorm |
|---|---|---|---|---|---|---|
| **Competencies** | CC, DC, PSL, CAE | MC, CC, DC, PSL, CAE, STEM. | STEM, CC, DC, CAE. | DC, CC. | DC, CC. | DC, STEM, CC. |
| **Skills** | Remember, understand, apply, analyse. | Remember, understand, apply, analyse, evaluate. | Remember, understand, apply, analyse, evaluate, create. | Remember, understand, apply. | Remember, understand, apply. | Remember, understand, apply, analyse, evaluate. |
| **Problem-solving conditions** | Productive reasoning, memory. | Productive reasoning, memory. | Productive reasoning, creativity. | Productive reasoning. | Productive reasoning. | Productive reasoning, creativity. |
| **Need for prior knowledge** | No | No | No | No | No | No |
| **Learning curve** | Medium difficulty | Medium difficulty | High difficulty | Low difficulty | Low difficulty | Medium-low difficulty |
| **Possibility of working in groups** | Yes | Yes | Yes | No | No | No |
| **Accessibility** | No | No | No | No | Yes | No |
| **Interdisciplinarity:** *combination of two or more academic disciplines* | No | Yes | Yes | No | No | Yes |
| **Possibility of teacher evaluation** | Yes, through correct answers. | Yes, through accumulated points. | Yes, through a life-saving score. | No | No | No |

*Note: Digital competence (DC); Citizenship competence (CC); Mathematical, Science, Technology and Engineering competence (STEM); Personal, social and learning to learn competence (PSL); Multilingualism competence (MC); Cultural awareness and expression competence (CAE)*

**Competing interests**

The contact author has declared that none of the authors has any competing interests.

**Acknowledgments**

This research was funded by MCIN/AEI/10.13039/501100011033 and European Union NextGenerationEU/ PRTR through grant number TED2021-129474B-I00 and Junta de Andalucía (Spain), research group HUM-613.





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
