# Peer review of "How to communicate and educate more effectively on natural risk issues to improve disaster risk management through serious games"

_EGUsphere, 2024_

## Author Comment (AC1)

| DIMENSIONS | ONLINE GAMES
[Figure]
 | MOBILE APPS
[Figure]
 |
|---|---|---|
| **Identification** | • Created by institutions
• English | • Created by individuals
• Geostorm: 11 languages |

The games permit faimiliarisation with natural hazards, causes and effects/consequences arise

| | **ONLINE GAMES** | **MOBILE APPS** |
|---|---|---|
| **Gameplay** | • Individual and multi-player mode
• Positive feedbacks in Build a Kit
• Positive and negative feedbacks in Disaster Master
• Positive feedbacks in Stop Disaster
• Presence of rewards systems
• Short duration (1 hour)
• High-Medium interaction | • Individual mode
• Positive feedbacks
• No presence of rewards systems
• Long duration (more than 1 hour)
• High interaction |

The games present mainly explorer players, which interact with the environment

| | **ONLINE GAMES** | **MOBILE APPS** |
|---|---|---|
| **Narrative** | • Medium-High relevance of narrative
• Build a Kit character: girl in a wheelchair and her friends in familiar situations
• Disaster master character: same as Buid a Kit in various scenarios
• Stop Disaster: no character, consists in creating buildings or defences | • Low-Medium relevance of narrative
• Geostorm character: male and femele adults as superheroes in space
• Earthquake relief rescue character: man and his dog in a city devastated by an earthquake
• Disaster Rescue Service character: rescuer in a city affected by floods |

The games present varied narratives, scenarios and characters, connection between present and future is possible

| | **ONLINE GAMES** | **MOBILE APPS** |
|---|---|---|
| **Content** | • Basic images with soft and cheerful colours
• Build a Kit and Disaster Master, link to the government website | • Geostorm uses complex images with dark colours
• Geostorm link to trailer of the film |

The games use alamist terms, do not provide information resources and do not present misconceptions or errors

| | **ONLINE GAMES** | **MOBILE APPS** |
|---|---|---|
| **Education** | • Cover most competences and Bloom´s Taxonomy and competences
• The Multiculturalism and Cultural Awareness and Expression Competences are addressed
• Medium-High dificculty | • Cover few competences
• Earthquake Relief Rescue and Disaster Rescue Service cover only the lowest cognitive levels of Bloom´sTaxonomy
• Low-Medium difficulty |

The games cover a broad spectrum of the cognitive levels of Bloom's Taxonomy and competences

**Online Game: Build a Kit, Disaster Master, Stop Disaster**
**Mobile Apps: Earthquake Relief Rescue, Disaster Rescue Service, Geostorm**

---

## Author Response (AR1)

**RESPONSE TO REVIEWER 1:**

Thank you for your constructive comments and suggestions. We believe addressing these comments will strengthen the paper and improve the message and key points we are trying to convey. Below, we respond to the specific comments, point by point and provide clarifications where necessary. We are confident that through this process we can improve the structure and effectiveness of the paper and communicate the results and conclusions more clearly.

Sincerely, Dr Mercedes Vázquez Vílchez (on behalf of all co-authors)

Thank you for putting together this piece of paper. I found the topic very interesting and has large potential to be investigated to contribute to the DRM field. It is definitely an interesting topic to research considering that more and more serious games are emerging to enhance DRM.

I have some comments you can consider to improve the manuscript since I believe it can be publishable if adjustments are considered. I will go by section to make my suggestions clear enough.

\*\* Thank you for this positive evaluation of our manuscript.

The abstract and introduction sections are clear and well-presented, but I would suggest that in the introduction you expand on any other participatory approaches that have been reported in the literature to better frame why the research team decided to focus on serious gaming. As it is framed so far you exclude other approaches which are also important such as participatory mapping, workshops, hackathons or any other in-person and online methods to engage different stakeholders.

\*\* In the introduction, we have expanded on other participatory approaches and we clarify because we focus on serious gaming. We have introduced some sentences: "Some important approaches in adaptive management incorporate the use of knowledge co-production, where scientists, politicians and other stakeholders work to interexchange, create and implement knowledge (van Kerkhoff and Lebel, 2006). In this sense, participatory mapping, workshop and hackathons are highlighted (e.g. Sullivan-Wiley et al., 2019; Trejo-Rangel et al., 2023; Macholl et al., 2023). These approaches introduce local knowledge of natural hazards into vulnerability evaluation, showing diverse vulnerabilities to natural hazards that are co-produced at local scales (Sullivan-Wiley et al., 2019). Experiential (Kolb, 1984) and transformative (Mezirow, 1995) learning remark the importance of action oriented to problem-solving, learning by-doing and how these processes create reflective thinking, theory generation and applications of knowledge, enabling behaviour change for adaptation to natural hazards (Sharpe, 2016; Lavell et al., 2012)" (line 36 to 44).

"Serious games allow users to visualise and explore phenomena that would otherwise be very difficult to experience as they enhance player immersion, and allow them to learn about the consequences of their actions at different points in time during a natural hazard (Solinska-Nowak et al., 2018; Heinzlef et al., 2024). In this way, serious games encourage experiential and transformative learning, as users try to reproduce a context as close to reality as possible that could allow to the players to enable behaviour change for adaptation and resilience to natural hazards (Villagra, 2023). The effectiveness of learning through serious games are also the immediate feedback and the emotional and sensorial experiences they provide, which is essential for learning to mitigate the effects of natural hazards (Solinska-Nowak et al., 2018; Heinzlef et al., 2024)" (line 49 to 56).

I liked how you explained serious gamin and their importance in the DRR in the theoretical framework, I think that is important considering that they have been used for some time already, however, you put together good references about them.

**\*\* Thank you for your kind words.**

In the material and methods section, the section starts mentioning that it is qualitative research, however, when the results are presented, there are quantitative findings based on the questionnaires that were applied. I rather find this research as a mixed-method approach that complements findings from different methods.

\*\* The approach to this study is qualitative. To avoid confusion, we have replaced the word "questionnaire" with "semi-structured interviews" (lines 200, 204, 291, 322).

For the content analysis, it is mentioned that the authors designed an instrument of analysis and evaluation. However, the results mostly focus on analysis and it is not very clear to me how it complements the focus group approach which for me should be a "focus group questionnaire.

\*\*Thank you for this comment. We have clarified this point. We have explained the content analysis in the text with the following paragraph and the Table B1 (Appendix) appears in the main text as Table 1:

"A content analysis of the selected games was then carried out, which is a research tool used for the quantifying and analysis of the presence, meaning and relationship between specific words, themes or concepts, and thus inferences to be made about the messages within the different analysis unit (e.g. websites, journals, games, etc). The type of content analysis in this study is conceptual analysis. In conceptual content analysis a concept is chosen for examination and the analysis involves quantifying and counting its presence. It is able to identify the intentions or communicative tendencies in the games; in turn, it describes the attitudes or behaviours that result from those communications, revealing patterns in communication content. The dimensions analysed in this study were those proposed by Ouariachi et al. (2017a). These authors adapted the theoretical Social Discourse of Video Games Analysis Model (Pérez-Latorre, 2010) in an analysis instrument for games about climate change through the Delphi method. The instrument presents 51 criteria or variables, which are analysed in regards to the messages within the texts, audio, static and dynamic images of games. These criteria were classified into five dimensions: identification (features that help identify and locate the game), gameplay (set of properties that describe the player's experience

within a given game system), narrative (discursive construction around a complex phenomenon), contents (analysis of the information and messages transmitted) and educational aspects (referring to competencies, skills and learning). These criteria are described in further detail in Table 1. The analysis of the games was carried out by the authors, who played the games and filled out a form containing the criteria mentioned above"(line 162 to 176).

For the focus group (questionnaire), it is also not very clear how the MAXQDA software was used for analysing this method. I would expect the platform to be used in the content analysis section and not here. I know it is mentioned in the results, but I missed the implementation of a different method in the focus group section to gather participants' inputs. For instance, interviews or even an online workshop or so. I am concerned about conducting a questionnaire with that number of participants (8 video game experts and 14 natural hazard experts), which was not clear until I went through the results and figured it out.

\*\*Thank you for this comment. We have clarified in the text that the semi-structured discussions were conducted online though open-ended questions. Two online semi-structured interviews were created using Google Forms, one addressed to natural hazards experts and earth science educators, and one addressed to video game experts (line 204-205). We have added the questions of the semi-structured interviews as supplementary materials (Tables S2 and S3). Once all the responses had been collected, codes were formulated based on their analysis using the program MAXQDA (2020) because this software allowed the authors to analyse the qualitative data collaboratively, create a common language in our codebook and reach consensus while benefiting from the unique perspectives of each team member (line 208 to 211).

I recognised that was very interesting to see how you selected the participants, but would be interesting to better understand how you approached them to answer the questionnaire.

\*\*We have introduced details about how we approached to experts. In the text we have explained that the survey was carried out from March 10 to May 20, 2022. After sharing the interviews online, the experts in natural hazards and educators in earth sciences (13 experts) responded during the first week. However, it was necessary to insist with 42 video game experts to collect just 8 responses (line 205 to 208).

In the results section, it is not a very friendly reader to keep mentioning tables that are pages ahead of the point you are reading. It is time-consuming and you get lost while you keep going to the table and coming back to the point where you stop reading. I strongly suggest that you try to condense the results of tables B1 to B6 into one figure. I am imagining a general description of these sections, followed by a figure, design, scheme, or something creative to picture what was found, and if the reader wants to get more details, can go to the detailed tables. Please allow yourself to think about how you would like to see that in a figure that represents the important information you have gathered already. I find this section very disconnected, and it does not give an overall picture of what was found.

\*\* We thank you for this comment. We have provided a new figure 1 that condenses the information presented in Tables B1 to B6. Tables B1 to B6 have been included as supplementary material (Tables S4 to S8).

Also, I suggest to bring the discussions to the results section. Then, results can be compared to what has been reported in the literature. Then, you can discuss the type of related hazard, and lack of inclusivity due to the predominant languages (especially English) even when the public which could benefit from using these games does not use them, and other aspects.

\*\* Based on your suggestion and that of the other reviewer, we have reorganized the text, and the paper includes separate sections for results, discussion and conclusion.

Regarding the focus group results, I would try to merge both groups that participated and bring only key elements that contribute to answering your question "What are the educational and communicative elements or characteristics that serious games should have to improve DRM?". You may notice that not all the answers from the questionnaire provide key information and you may need to prioritise what helps to respond to the question. If you decide to present the previous section (content analysis) on how serious games are in a more condensed way, you could take advantage of that and highlight how they "should improve" with these findings. I can see that the data collected in the content analysis could help to build a current picture and the information collected in questionnaires would be the transformed version of that.

\*\*Thank you for this comment. We have eliminated those answers of the questionnaires that fail to provide key information for answering the research question "What are the educational and communicative elements or characteristics that serious games should have to improve DRM?". We have deleted the responses from videogame experts on curiosity and chance (line 291 to 298 in older version) and the corresponding older Figure 2. In this way, we have highlighted those key answers directly related to the summary figure of content analysis of the games analysed (new figure 1).

Some charts and tables look a bit unnecessary since they do not say so much and take up a large space in the document. Please reconsider which of them are useful and how would be a better way to present more connected results where discussion can be included. I do see that you have collected valuable information, I just do not get how the results are organised, I believe this section should be better presented.

\*\*Thank you for this comment. We have reorganised the information presented. We have incorporated the key information of the older Figures 1, 3 and 4 in Tables 2 and 3 (categories of the expert responses) and, therefore, these older figures have been deleted.

I suggest leaving the conclusions as a separate section without merging it with the discussions since I see a lot more potential to include discussions in the results section.

 $\ensuremath{^{**}}\xspace$  we now present the results, discussion and conclusions in separate sections in the paper.

**RESPONSE TO REVIEWER 2**

Thank you for your constructive comments and suggestions. We believe addressing these comments will strengthen the paper and improve the message and key points we are trying to convey. Below, we respond to the specific comments, point by point and provide clarifications where necessary. We are confident that through this process we can improve the structure and effectiveness of the paper and communicate the results and conclusions more clearly.

Sincerely, Dr Mercedes Vázquez Vílchez (on behalf of all co-authors)

This paper examines the potential benefits of serious games for enhancing disaster risk management. The researchers employed a methodological triangulation approach, which involved content analysis, focus group discussions with experts and literature review. The findings show that online games are more successful in meeting experts' narrative criteria and have higher educational value, while mobile apps focus more on interaction, increasing engagement but lacking educational depth. The study also highlights a lack of attention to multiculturalism, diversity, and gender issues. Thank you for your manuscript. The research questions posed are interesting and the research, per se, is relevant to the research area. However, the paper's structure and the presentation of the information lack clarity and the grammar used is not at a level for publication.

- \*\* We thank the reviewer for the careful review. We agree that the paper will greatly benefit from modifying the structure, improving the presentation of the information, and enhancing the grammar.
- The grammar in the paper is seriously lacking throughout major parts of the paper. Therefore, please have the paper be proofread by a native speaker.
- \*\* A native speaker has revised the paper.
- The introduction would benefit from a clearer delineation of the research gap(s) and how this paper specifically is trying to answer it/them.
- \*\*Thank you for this comment. We have clarified the research gaps in the introduction section. The revised text includes the following sentences "Few works address the influence of video games on the tendency of players to prepare for natural hazards (Tanes and Cho, 2013; Tanes, 2017; Safran et al, 2024). Attention is drawn to the lack of solid scientific evidence of the potential of serious games, with challenges remaining for the development of more detailed studies to test and demonstrate the effectiveness of serious games for DRM education (Weyrich et al., 2021; Safran et al., 2024)" (line 62 to 86).
- I would appreciate it if the research questions was mentioned in the introduction and then specifically taken up again in the discussion part and answered in more detail.

\*\*Thank you for this comment. We have specified the research question in the introduction. Now, the following paragraph appears in the text: "This paper aims to explore the potential of serious games for improving DRM. The main research question it raises is: How can we educate and communicate more effectively about natural hazards through serious games? To this end, the following research subquestions are posed: (a) How do serious games communicate and educate on issues related to natural hazards? and (b) Which educational and communicative elements or characteristics should serious games have to improve DRM? This work will follow a methodology consisting of triangulation-based qualitative research. The research method includes a content analysis of selected serious games applied to DRM, a focus group of experts and a literature review (the constructs of the investigation were determined with the help of introductory literature)" (line 87 to 93). Also we have indicated the research question in the discussion to provide a more detailed answer (line 734 and 761).

-Explain in more detail how the information in the results section were obtained. Adding the questionnaire to the appendix could be helpful to remedy that and help readers understand what the questionnaire looked like.

\*\*The questions of the semi-structured interviews have been added as supplementary materials (Table S2 and S3).

-consider if all of the figures presented in the results section are actually necessary or if the information presented in them could be condensed into one or two figure(s). Also rethink how the information in the tables is presented (see specific comment below).

\*\*We agree that not all figures presented are necessary. We have incorporated the key information from Figures 1, 3, and 4 into Tables 2 and 3 (expert response categories) and, therefore, these figures have been deleted. Figure 2 has also been eliminated based on feedback from another reviewer.

-Some parts of the results section rather belongs to the discussion and conclusion chapter, i.e. leave all interpretation of the results to the discussion. Some, but by far not all, of the instances are commented on in the specific comments (see specific comment below).

\*\*Thank you for this comment. The interpretations have been moved from the results to the discussion and conclusion sections.

- I would separate the discussion and conclusion section so as to make it more reader-friendly.

\*\*We appreciate the suggestion and have separated the discussion and conclusions.

- the paper could benefit from a clearer use of terminology = unify the use of i.e. natural risks, natural hazards, natural hazards-related disaster.

\*\* We have unified the terminology using only natural hazards.

- refrain from using evaluating language if it is not language used by the experts (and if the latter, use direct quotes).

\*\*We thank you for this comment. We use the language used by the experts. However, we acknowledge the potential confusion and we have added two tables with responses of the experts as supplementary material (Tables S9 and S10).

**Specific comments:**

Section 2 (Theoretical framework):

Line 99: add percentage for the prevalence of storms.

\*\*We have added 34% as a percentage for the prevalence of storms (line 151).

Section 3 (Materials and methods):

Line 135: the age range is unclear – if the game is geared at young people without giving an age upper limit, it should rather be argued that it is for young and adult people.

\*\*We have clarified that the games analysed are aimed at young people and adults (age 12 and upwards) (line 192).

Line 137: it would be helpful for the reader to know on the basis of which criteria this selection was made.

\*\*We have remarked that according to the criteria explained previously, in this phase, from 17 digital games, a total of 6 games were selected: 3 mobile apps (Earthquake Relief Rescue, Geostorm and Disaster Rescue Service) and 3 online games (Build a kit, Disaster Master and Stop Disasters) (line 193 to 195).

Line 140ff: describe in more detail what the content analysis looked like with games as objects of the study. How was content extracted from the games, which aspects of the structure of the games were analyzed (i.e. visual, audio, text, etc.) and how. How did you deal with the interactive aspect of the games, i.e. different outcomes of the games depending on the players' decisions, etc.?

\*\* Thank you for this comment. We have clarified this point. We have explained the content analysis in the text with the following paragraph and the Table B1 (older Appendix) appears in the main text as Table 1:

"A content analysis of the selected games was then carried out, which is a research tool used for the quantifying and analysis of the presence, meaning and relationship between specific words, themes or concepts, and thus inferences to be made about the messages within the different analysis unit (e.g. websites, journals, games, etc). The type of content analysis in this study is conceptual analysis. In conceptual content analysis a concept is chosen for examination and the analysis involves quantifying and counting its presence. It is able to identify the intentions or communicative tendencies in the games; in turn, it describes the attitudes or behaviours that result from those communications, revealing patterns in communication content. The dimensions analysed in this study were those proposed by Ouariachi et al. (2017a). These authors adapted the theoretical Social

Discourse of Video Games Analysis Model (Pérez-Latorre, 2010) in an analysis instrument for games about climate change through the Delphi method. The instrument presents 51 criteria or variables, which are analysed in regards to the messages within the texts, audio, static and dynamic images of games. These criteria were classified into five dimensions: identification (features that help identify and locate the game), gameplay (set of properties that describe the player's experience within a given game system), narrative (discursive construction around a complex phenomenon), contents (analysis of the information and messages transmitted) and educational aspects (referring to competencies, skills and learning). These criteria are described in further detail in Table 1. The analysis of the games was carried out by the authors, who played the games and filled out a form containing the criteria mentioned above" (line 203 to 217).

Line 144: "First" is not needed if there are no other arguments listed ("second", "third").

\*\*We thank you for this comment. We have modified this paragraph and the first word has been deleted.

Line 147: fix reference (only year in brackets).

\*\*We have corrected this reference (line 168).

Line 152: close bracket.

\*\* We have closed the brackets (line 180).

Lines 163-165: Rewrite accordingly: This study involved individuals with at least 5 years of professional or experiential knowledge of the research topic, constituting an informed panel and thereby justifying the use of the title "experts" (Mullen, 2003).

\*\*Thank you for this comment. We have properly rewritten this sentence (line 278 to 279).

Line 170: include an explanation of how these participants are experts in natural hazards, i.e. are they researchers, practitioners, stakeholders...

\*\*We have added the following explanation to the paper: "As regards the surveys of experts in natural hazards, they present different natural hazard backgrounds (climate change, floods, earthquakes, volcanoes and mass movement) and professions (researchers, emergencies experts, a politician and the partner of a consulting company in urban and territorial planning)" (line 283 to 286).

Line 179: add citation for MAXQDA program. Also, it is unclear what the program was used for.

\*\*Thank you for this comment. We have included the correct reference; MAXQDA (2020) appears in the text and we explain that this qualitative data analysis software allowed the authors to analyse the data collaboratively, create a common language

in our codebook and reach consensus while benefiting from the unique perspectives of each team member (line 316 to 319).

**Section 4 - Results:**

Clarify how this information was obtained. Were the following questions posed as open questions, was there a selection of answers given and could experts add own ideas, ...? As mentioned above, adding the appendix to the paper helps readers understand how this information was elicited. This applies to all other questions in the following tables in the result section.

\*\*Thank you for this comment. The semi-structured discussions were conducted online though open-ended questions (line 308). We have added the questions of these as supplementary materials (Table S2 and S3).

Rethink the use of figures (see comment above) and consider condensing some figures into one. Also, the tables do not present information in a very reader-friendly way, consider rephrasing the items shown so it is more clear to the readers what they mean.

\*\*We have incorporated the key information from Figures 1, 3, and 4 into Tables 2 and 3 (expert response categories) and, therefore, these figures have been deleted. Figure 2 has also been eliminated based on feedback from another reviewer. We have provided a figure (new Figure 1) that condenses the information presented in Tables B1 to B6. Tables from B1 to B6 have been included as supplementary material (Tables S4 to S8). In order to improve the understanding of the tables some categories have been rephrased (new Table 3: "Inequalities determinant" and "No dependent on vulnerability").

Line 188: To me, there seems to be no point in highlighting Hungarian and Turkish as available languages for the paper at hand, so delete and rather than pointing out specific languages maybe state the number of languages in which the game is available.

\*\*We have specified that the Geostorm game gives the option of 11 languages (line 382).

Line 221: add why the term "natural risk" in particular is one you looked out for. Also I would, again, argue that this is content for the discussion section of the paper.

\*We thank you for this comment. We have deleted the following sentence: "We did not find the term "natural risk", but rather "natural catastrophe" or "natural disaster", because in the text it is enough clear that the games use rather alarmist terms such as "emergency", "catastrophe" and "disaster".

Line 221-222: explain in more detail – why is this an interesting result and what constitutes a complex concept (consider also giving examples). Again, this would be content for the discussion section.

\*\*We agree, and this sentence has been moved to discussion section. We have added examples of complex concepts such as earthquake epicentre, floodplain, etc (line 759 to 760).

Line 231: Consider moving this whole subsection (4.1.4) to the beginning of the chapter. Painting a picture of the games' setup could help readers in getting an understanding of the games and differentiating them better.

\*\*We have relocated the gameplay subsection to the beginning of the chapter. We have provided a figure (new Figure 1) that condenses the information presented from Tables B1 to B6, including Table B5 (game play dimension).

Line 233: use "in multi-player mode" instead of "collective way".

\*\*We agree, we use multi-player mode (example Table 1 and Table S5).

Line 238: make the link clearer as to why the online games offer educational advantages over the mobile ones. What characterizes the mobile ones in comparison?.

\*\*The sentence: "Therefore, these three online games offer further educational advantages over the mobile app examples" has been deleted. This idea has been integrated into the conclusion section (line 465-469).

Line 243: Clarify what you mean by "the other games might also be played in an hour".

\*\*Thank you for this comment. The sentence has been modified as: "The other games present levels that might also be played in one hour" (line 400).

Line 245: Clarify what the difference is between positive feedback and a reward system.

\*\* We have clarified this point through the following addition to the text: "Positive feedback, though messages the player receives in light of certain actions, are abundant in the games, and we only found a reward system in the online examples (line 401 to 402)". In addition, this point is extensively explained in Table B1 (Table 1 in new version).

Line 250: Explain what Bloom's Taxonomy is and how this is relevant here.

\*\*Thank you for this comment. We have explained that Bloom's Taxonomy consists of a hierarchical structure of objectives or levels that allow educators to evaluate the learning process of students; it is also a useful starting point for designing activities to achieve meaningful and lifelong learning. Accordingly, the evaluation criteria related to "Remember" and "Understand" are classified as "Basic"; the criteria related to "Apply" and "Analyse" are catalogued as "Optimal"; and the criteria related to "Evaluate" and "Create" are classified as "Desirable". Taking into account these levels, the most complete are Stop Disasters, Disaster Master and Geostorm which cover all of them; however, Earthquake Relief Rescue and Disaster Rescue

Service are the two games that cover only the Basic and Optimal learning levels (line 458 to 465).

*Line 258f: Clarify if only online games dealt with these dimensions.*

\*\*We have clarified that the Multiculturalism and Cultural Awareness and Expression Competences are addressed only by the online games (Build a kit, Disaster Master and Stop Disasters) (line 471 to 472).

Line 266ff/Table 1: clarify what "Always" means in "reward systems" section, especially since this is not referred to in the text. Also consider renaming this item as it is too unspecific.

\*\*Based on the other reviewer, this category has been eliminated from Table 1 (Table 2 in revised version), because it does not contribute towards answering the research question. In addition, the idea expressed by the experts in the recommended/satisfactory category is too similar to the Always category.

Line 266ff/Table 1: clarify why progress and motivation is handled as one category. Consider renaming the third category to "motivate" in order to make terminology uniform (i.e. all verb infinitives or the gerund). Do this for the entire table.

\*\*According to the suggestion of the other reviewer, these categories from the dimension level have been eliminated from Table 1 (Table 2 in the new version), as they do not contribute towards answering the research question.

Line 266ff/Table 1: the last 3 subcategories in the "duration" section are confusing to me as their connection to the theme of duration only makes sense after reading the text. This, to me, raises the question if the information should rather be presented in an answer-like style, i.e. "game should not take more than an hour" "game should take up to several hours", "game length can be variable if narrative is employed", etc. Such rephrasing could help readers understand the subcategories better.

\*\*Based on the suggestion of the other reviewer, the categories 'narrative' and 'engage' have been eliminated from Table 1 (Table 2 in new version) as they do not contribute towards answering the research question. We have added new tables with examples of expert responses in the supplementary materials to improve the understanding of the remaining categories (Tables S9 and S10).

Line 280: 46% does not appear in the table. Check if numbers in text and table align.

\*\*We thank you for this comment. We have corrected the data in new Table 2.

Line 291: Figure 1 is not referred to in the text and, even if it were, seems unnecessary – information depicted does not need its own figure.

\*\*Figure 1 has been deleted and the data appear in the new Table 2.

Line 293: "All of the experts agree" does not fit to the number in the cited figure (figure only shows 62% for curiosity).

\*\*According to the other reviewer, we have eliminated those answers of the semistructured interviews that do not provide key information towards answering the research question: What are the educational and communicative elements or characteristics that serious games should have to improve DRM? We have deleted the responses from videogame experts on curiosity and chance (line 291 to 298 in older version)) and we have deleted the corresponding Figure 2. In this way, we have highlighted those key answers directly related to the content analysis of the games evaluated.

Lines 295ff: the last to sentences of the paragraph seem to repeat the same information.

\*\*As we mentioned in the previous point, we have deleted the responses from videogame experts on curiosity and chance (line 291 to 298 in older version).

Line 301: Unsure whether this conclusion can be drawn from merely 7% and 21% of responses.

\*\*We thank you for this comment. We have deleted the categories fun and engage, corresponding to 7% and 21% of the level of interaction dimension. We have included the level of interaction dimension with only one category (narrative) in the new Table 2.

*Line 302f: This is an interpretation, move this part to the discussion section.*

\*\*We appreciate the suggestion and we have modified this sentence as: "However, the code most represented in this question is narrative, since in a serious game with strong narrative the interaction can be lower and they can have a great impact." We have added two tables in supplementary materials showing the experts' answers.

Table 2: there is a typo in "multiculturalis".

\*\*We have corrected this typo.

Table 2: change phrasing of "normal person" – avoid evaluative language if it is not language explicitly used by the experts. If that were the case, use direct quotes.

\*\*We thank you for this comment. We do not use evaluative language; instead, we use the language explicitly employed by the experts. We will thoroughly review the results to relocate any sentences that include the authors' interpretations. We have added two tables with direct quotes from experts to help readers understand the results (Tables S9 and 10).

Figure 5: figure is cut off on the left.

\*\*We have corrected this in the revised version.

Line 363f: cut "wonderful" and "great" - avoid evaluative language if it is not language explicitly used by the experts. If that were the case, use direct quotes.

\*\*We appreciate the suggestion and, in this case, we have deleted these words from the text to avoid evaluative language.

Section 5 (discussion and conclusion):

Move all interpretation of the obtained data currently found in the results section here and consider splitting discussion and conclusion into two separate chapters.

\*\*We have moved all interpretations from the results section to a new discussion section. We have provided a separate section for the conclusion.

Lines 402ff: rephrase this sentence, as it is not clear which of the aspects named are those of "secondary presence".

\*\*We have rephrased this sentence. The new version includes: "These desirable characteristics are: exploratory characters in a cooperative dynamic, with simple and non-technical narratives, based on academic sources, multiculturalism, consideration of diversity and gender, fun, short games, constructive feedback system, while the rewards could have a secondary presence (lines 456 to 459)."

Section 6 (limits and recommendation for future studies):

This section requires more detail and clarity, i.e. what aspects of the methodological approach of triangulation helps in preventing a subjective interpretation of the results? Clarify what you mean by "seeking consensus". Give more detailed examples of aspects to be studies for future research.

\*\* We have rewritten this section. In the new version of the paper, it appears as:

"The limitations of this study are related to the subjective nature of qualitative analyses, due to their focus on interpreting meaning and the meaning-making process. Consequently, a methodical triangulation approach (content analysis of selected serious games, focus group with experts and literature review) was used to minimize this limitation (Ouariachi et al., 2017b)" (line 471to 473).

On the other hand, in section 3.2 Focus group with experts (3. Materials and methods) we will clarify that the authors analysed the data through several rounds, seeking consensus to establish codes (line 208 to 211) and the older sentence "In addition, the authors analysed the data through several rounds, seeking consensus among the established codes will be deleted.

"This study is exploratory in nature, thus we encourage researchers to delve deeper into how videogames can enhance DRM. Further research could improve our understanding of how specific narrative and gameplay elements (e.g. collaborative or competitive, duration of game and feedback and rewards systems), mechanics (e.g., mission achievement, creating new resources, discovering clues), and characters (e.g. different player roles, character characteristics, selectable avatars) in disaster-related video games improve DRM education. In this sense, subsequent studies could concentrate on validating the effectiveness of the proposed features

in enhancing behaviour change of citizens for improving adaptation and resilience to natural hazards" (line 474 to 480).

**Appendix:**

Table B3 – Disaster Master/global history: rearrange this paragraph, moving the last sentence to the front ("the overarching framework ...").

\*\*We thank you for this comment. We have rearranged this paragraph according to your suggestion (see Table S6).

Table B3 – Earthquake Relief Rescue+/Representation of character: cut out "but in a very impersonal manner" – avoid evaluating language.

\*\*We have removed "but in a very impersonal manner" (see Table S6).

*Table B6 – Competencies: include a footnote explaining the abbreviations.*

\*\*We appreciate the suggestion; however, a footnote explaining the abbreviations was included (see Table S8).

Table B6 – Learning curve: consider replacing "learning curve" with "level of difficulty".

\*We have replaced "learning curve" with "level of difficulty" (see Table S8).

Table B6 – Accessibility: Explain what is meant by accessibility.

\*We thank you for this comment. We have explained that accessibility refers to possibilities for people with functional diversity (see Table S8).

---

## Referee Report (RR1)

Thank you for your work in improving the manuscript. As stated previously, the research questions posed are engaging, and the study contributes to the field. Although the manuscript has seen significant improvement, there are still areas that need attention.

**General Comments:**

The grammar and spelling still need considerable improvement and are not yet at a publishable level. This includes correcting typos, grammar, and sentence structures; using appropriate linking words; ensuring consistent comma usage; removing double periods (lines 164 and 309); and unifying the capitalization of game titles (e.g., line 289). Also, check Figure 1 for typos.

Write out numbers below ten and apply this rule consistently. For example, in line 191, "twenty-two" is written out, while "5" in the line above is written as an Arabic numeral.

**Methodology:**

I do not support your claim that methodological triangulation was used. What is described as a literature review appears to be primarily the references used in the discussion and results sections to support arguments, rather than an independent research method. Since the study does not employ a systematic or structured approach—such as a systematic review, scoping review, or meta-analysis—to synthesize existing knowledge, it appears that only two research methods were applied (content analysis and expert focus groups). Therefore, it would be more accurate either to remove this reference to methodological triangulation, explicitly conduct a structured literature review, or adjust the wording to clarify that data triangulation (i.e., incorporating different sources of information on the same topic) was used instead.

I recommend that you eliminate sentences 202–204: "For this reason, we endeavored to include specific questions, eliminating those that were similar, avoiding questions that were too open-ended, and focusing on those that allowed for a relevant response to the topic of study." In my opinion, this sentence does not add meaningful information.

**Results:**

**Paragraph 259–262:** Be cautious when stating that the games exhibit a "great diversity" of characters based on the descriptions in the text. If the race and ethnicity of the main characters are not specified, the claim may not be well-supported. Additionally, simply allowing players to choose between a male and female character does not constitute "great diversity" and fails to acknowledge non-binary identities. It would be more accurate to state that the games include *some* representation—for instance, by featuring a protagonist with a mobility impairment or including characters from different backgrounds. This is especially relevant given your statement in line 408: "Few games consider multicultural and inclusivity aspects."

**Section 4.3.XXX:** Ensure that it is clear to the reader that the points listed in the first paragraphs of each subchapter reflect the opinions of the experts rather than universally accepted facts. Additionally, improve readability by incorporating linking words.

Line 270: "Three out of six games" is not the majority.

**Figure 2:** It is unclear why Figure 2 stands alone rather than being included in one of the expert response tables. Please delete the figure and integrate the information into Table 3.

**Tables 2 and 3:** The table descriptions should be made more informative to better convey the content they present. Ideally, a table title should clearly state what the table contains while providing a brief summary of its contents. The column title "Importance" is ambiguous, as it suggests a qualitative measure rather than a numerical representation of expert agreement. A more precise title would be "Expert Agreement (%)" or "Proportion of Experts (%)." Additionally, consider whether Tables 2 and 3 could be combined to save space by presenting them side by side.

I also do not fully understand how you arrived at the response numbers in Tables 2 and 3, given that Table S9 appears to display all expert responses. For example, how did you determine that "Character – Socializer" received five responses and "Character – Explorer" received three when Table S9 suggests only two responses were recorded? Please clarify.

**Discussion:**

Thank you for restructuring this section; it is a significant improvement. However, this section still lacks clarity. Some parts focus too heavily on reiterating the research results rather than analyzing their implications and interpretations. The discussion should clearly summarize the key findings in relation to the research questions and critically reflect on them.

Additionally, the discussion lacks an in-depth comparison to existing research, which would strengthen your arguments. A few specific areas that could be expanded include:

- The debate over whether a catastrophic or non-alarmist tone is more effective is interesting but remains inconclusive. Rather than simply presenting both perspectives, consider providing insights into how game designers could balance these approaches effectively.
- While you discuss how serious games engage players, you could further explore
  whether they improve understanding of scientific concepts related to natural hazards.
  The mention of Bloom's Taxonomy is useful but could be expanded by specifying
  which cognitive skills these games enhance the most.
- The argument that excessive rewards can be distracting (Chou, 2015) is valid, but it would be helpful to include practical recommendations on how to structure rewards to enhance learning rather than just entertainment.
- You mention collective efficacy and community engagement but do not explore in depth how these elements could be effectively integrated into game mechanics. Providing concrete examples from existing games would strengthen this argument.

**Conclusion:**

This section would benefit from deeper reflections on what your research has achieved. The statement that the study provides "new insights" is somewhat vague—be more specific about what is novel in your findings compared to previous research.

While you acknowledge the limitations of qualitative analysis, it would be useful to suggest how future studies could address these limitations more concretely (e.g., through experimental studies to measure behavioral changes).

The recommendations for future research are relevant but somewhat broad. Instead of simply suggesting further studies on different game elements, propose specific research questions or methodologies that could build upon your findings.

---

## Referee Report (RR2)

The paper at hand addresses the question of how serious games can be implemented to improve disaster risk management using a qualitative methodological approach combining content analysis and an online survey of experts. The manuscript has improved considerably and the feedback given in the previous review round has been addressed. There are still some issues with incorrect grammar and some unclarity regarding the content, which warrant minor revisions.

**Content:**

- a) Throughout the manuscript, you keep using different terms for your process with experts ("semi-structured discussions", "online semi-structured interviews", "survey", "expert-led focus groups", "focus group of experts"), which is confusing. In my understanding, you sent out an online survey with open-ended questions, to which 8 experts then replied. If this is correct, please make this clear in throughout the manuscript. You may also add a comment about the rather small size of the expert group in the limitations part of the conclusion and what it means for the reliability of the study.
- b) Line 117: Why is the reference year 2015? Which hazard has been the most common since then? 2015 was 10 years ago, so saying in the last 20 years floods have been the most common only makes it the most common for half of the timeframe what about the other half? Add information about what has been the most common hazard since 2015
- c) Figure 1: under "mobile apps" and "gameplay": add which game is referred to when you talk about "no reward systems" and "provide positive feedback". Under "narrative" and "online games": change last line to "No character; game focuses on constructing buildings or defences". Under "Content" in the blue box: use a comma after "terms", i.e. "use alarmist terms, do not provide ..."
- d) Line 283: The European education curriculum comes quite out of the blue maybe add a sentence or two on what the curriculum states and its aims and why you found it worthwhile to include it in your analysis

**Grammatical errors:**

Lines 23 to 25 – this sentence is unclear and needs rewriting.

Line 39 – interchange or exchange

Line 53 – could allow the players (eliminate the to)

Line 106 – grammatically incorrect, change to: "Considering their characteristics, there is a wide variety of genres and formats such as simulations, which replicate aspects of real or fictional realities and adventures, in which users solve challenges by interacting with people or the environment in a non-confrontational manner."

Line 126 - exchange "achieve" with "reach"

Line 131 – exchange "destined to" with "intended for"

Line 135 – exchange "DRR is the lack" with "DRR being the lack"

Line 158 – change to: "A content analysis of the selected games was then carried out. This is a research method used to quantify and analyze the presence, meaning, and relationships of specific words, themes, or concepts, allowing for inferences to be made about the messages within different units of analysis"

Line 162 – eliminate the "in turn" (i.e. have the sentence go like this: "... tendencies in the games and describes the attitudes ...")

Line 164 – change to "These authors adapted the theoretical Social Discourse of Video Games Analysis Model (Pérez-Latorre, 2010) into an analytical instrument for games about climate change using the Delphi method."

Line 221 – add "could be used in both"

Line 225 – change to: "Finally, in Stop Disasters, decisions such as which buildings to construct or improve, where to build hospitals, or which barriers to build against natural hazards can also involve teamwork."

Line 230: eliminate "in the content"

Line 231: Change to: "Build a Kit is the only game that can likely be completed within an hour, while Disaster Master may also be finished in that time, depending on the students' age and comprehension level."

Line 234: eliminate the comma after "actions"

Line 253: change to: "As it is based on a fictional film, Geostorm leans more toward fantasy and is less realistic than other games, such as Disaster Master."

Line 278: exchange "however" with "meanwhile"

Line 281: Eliminate the "likewise"

Line 299: change to: "... since natural phenomena lend themselves well to interaction with the environment."

Line 318 – I am unsure what that sentence means. If this is what you want to say, please change to: "However, the most prominent element in this context is narrative, since serious games with a strong narrative may have less interaction but can still deliver a powerful impact."

Line 343: "since livelihoods, housing, and access to training and information are often unequal and more susceptible to natural hazards"

Line 346: change to "training in prevention and vulnerability reduction"

Line 368: change to "integration of findings" or "synthesis of findings"

Line 369: change to "In this section, the findings of the qualitative content analysis and the expert focus group were compared and further summarized"

Line 373: change to "multiculturalism" instead of "multicultural"

Line 468: there is a word missing

480: "Reward system" - eliminate the s in "Rewards"

---

## Author Response (AR2)

We sincerely thank Reviewer 2 for the thoughtful and detailed feedback. We appreciate your recognition of the improvements made to the manuscript and your valuable recommendations for further refinement. Below we provide a point-by-point response to each of your comments. All page and line numbers provided below refer to the revised version of the manuscript with tracked changes.

**General Comments**

Reviewer comment:

The grammar and spelling still need considerable improvement and are not yet at a publishable level. This includes correcting typos, grammar, and sentence structures; using appropriate linking words; ensuring consistent comma usage; removing double periods (lines 164 and 309); and unifying the capitalization of game titles (e.g., line 289). Also, check Figure 1 for typos. Write out numbers below ten and apply this rule consistently. For example, in line 191, "twenty-two" is written out, while "5" in the line above is written as an Arabic numeral.

**Author response:**

Thank you for this observation. We have carried out a comprehensive revision of the manuscript to correct grammatical and spelling issues, ensure consistent formatting (including punctuation and capitalization), and improve sentence flow and readability. Specific corrections were made to lines 164, 289, and 309. Figure 1 has also been corrected for typographical errors. We have also standardized the use of numbers, writing out those below ten consistently throughout the text. However, in cases where a sentence begins with a number, we have followed the journal's editorial guidelines and spelled out the number, as this is the recommended practice for formal academic writing (page 7, line 282).

**Methodology**

Reviewer comment:

I do not support your claim that methodological triangulation was used. What is described as a literature review appears to be primarily the references used in the discussion and results sections to support arguments, rather than an independent research method. Since the study does not employ a systematic or structured approach—such as a systematic review, scoping review, or meta-analysis—to synthesize existing knowledge, it appears that only two research methods were applied (content analysis and expert focus groups). Therefore, it would be more accurate either to remove this reference to methodological triangulation, explicitly conduct a structured literature review, or adjust the wording to clarify that data triangulation (i.e., incorporating different sources of information on the same topic) was used instead.

**Author response:**

We appreciate this clarification. The reference to "methodological triangulation" has been removed from the manuscript. We have clarified that the study employed two main qualitative research methods (expert interviews and content analysis) in Abstract (page 1, line 11-12), Introduction (page 3, line 126-127) and Materials and methods (page 5, line 207), and Conclusions (page 19, line 830).

**Reviewer comment:**

I recommend that you eliminate sentences 202–204: "For this reason, we endeavored to include specific questions, eliminating those that were similar, avoiding questions that were too open-ended, and focusing on those that allowed for a relevant response to the topic of study." In my opinion, this sentence does not add meaningful information

**Author response:**

As suggested, these sentences have been removed from the manuscript.

**Results**

**Reviewer comment:**

Paragraph 259–262: Be cautious when stating that the games exhibit a "great diversity" of characters based on the descriptions in the text. If the race and ethnicity of the main characters are not specified, the claim may not be well-supported. Additionally, simply allowing players to choose between a male and female character does not constitute "great diversity" and fails to acknowledge non-binary identities. It would be more accurate to state that the games include some representation—for instance, by featuring a protagonist with a mobility impairment or including characters from different backgrounds. This is especially relevant given your statement in line 408: "Few games consider multicultural and inclusivity aspects."

**Author response:**

We thank the reviewer for this valuable comment. In response, we have revised the paragraph to provide a more accurate description of character representation. Specifically, we replaced the original statement "highlight the great diversity present in the games analysed" with "acknowledge the limited representation present in the games analysed", and clarified the limitations by explicitly mentioning the lack of broader representation regarding race, ethnicity, and non-binary identities (page 10, lines 397-402).

**Reviewer comment:**

**Section 4.3.XXX:** Ensure that it is clear to the reader that the points listed in the first paragraphs of each subchapter reflect the opinions of the experts rather than universally accepted facts. Additionally, improve readability by incorporating linking words.

**Author response:**

We have revised the introductory paragraph of each subsection in Section 4.3 to explicitly indicate that the points presented reflect expert opinions (page 14, line 578, and page 15 lines 626 and 637). We have also added linking words to improve overall readability and cohesion.

**Reviewer comment:**

*Line 270:* "Three out of six games" is not the majority.

**Author response:**

Thank you for pointing this out. The sentence has been corrected to avoid the inaccurate use of the term "majority" (page 11, line 406).

**Reviewer comment:**

**Figure 2:** It is unclear why Figure 2 stands alone rather than being included in one of the expert response tables. Please delete the figure and integrate the information into Table 3.

**Author response:**

We have removed Figure 2 and integrated its information into an updated version of Table 3, as suggested.

**Reviewer comment:**

**Tables 2 and 3:** The table descriptions should be made more informative to better convey the content they present. Ideally, a table title should clearly state what the table contains while providing a brief summary of its contents. The column title "Importance" is ambiguous, as it suggests a qualitative measure rather than a numerical representation of expert agreement. A more precise title would be "Expert Agreement (%)" or "Proportion of Experts (%)." Additionally, consider whether Tables 2 and 3 could be combined to save space by presenting them side by side.

**Author response:**

We thank the reviewer for this valuable suggestion. In response, we have revised the titles and descriptions of Tables 2 and 3 to provide more informative summaries. The column previously titled "Importance" has been renamed "Expert Agreement (%)" to accurately reflect the data.

Although the tables have not been merged, they are now displayed side by side to improve space efficiency and facilitate comparison without compromising clarity.

**Reviewer comment:**

I also do not fully understand how you arrived at the response numbers in Tables 2 and 3, given that Table S9 appears to display all expert responses. For example, how did you determine that "Character – Socializer" received five responses and "Character – Explorer" received three when Table S9 suggests only two responses were recorded? Please clarify.

**Author response:**

We have clarified in the text that Tables 2 and 3 are based on the full set of coded responses obtained through thematic analysis, which included semantic grouping of similar expert answers. Table S9 and S10 presents selected examples and does not include all responses. This clarification has been added in Results section (page 11, line 452-453) and in description of Table S9 and S10 (see Supplementary materials).

**Discussion**

**Reviewer comment:**

Thank you for restructuring this section; it is a significant improvement. However, this section still lacks clarity. Some parts focus too heavily on reiterating the research results rather than analyzing their implications and interpretations. The discussion should clearly summarize the key findings in relation to the research questions and critically reflect on them.

**Author response:**

We sincerely thank the reviewer for their valuable feedback and for highlighting the need to improve the clarity of the discussion section. In response, we have carefully revised this section to ensure it clearly addresses the research questions and provides a more critical interpretation of the results. Specifically, we have:

- Structured the Section 5.1 to directly respond to the first research question, focusing exclusively on how serious games communicate and educate about natural hazards.
- Revised the Section 5.2 to address the second research question, clearly outlining the recommended educational and communicative elements that serious games should incorporate to improve DRM.

To support these revisions, we have also incorporated several new references that reinforce the theoretical and empirical grounding of the discussion. These have been added to the updated References section (page 21, lines 1352-1354; page 22, lines 1402-1404; page 22, lines 1405-1406).

**Reviewer comment:**

Additionally, the discussion lacks an in-depth comparison to existing research, which would strengthen your arguments. A few specific areas that could be expanded include:

• The debate over whether a catastrophic or non-alarmist tone is more effective is interesting but remains inconclusive. Rather than simply presenting both perspectives, consider providing insights into how game designers could balance these approaches effectively.

**Author response:**

Thank you for this valuable suggestion. In response, we have expanded the relevant section to propose concrete strategies for balancing emotional intensity with player empowerment in game design. The revised text now discusses how serious games might open with immersive, high-stakes scenarios to capture attention and elevate perceived risk, followed by interactive phases that reinforce decision-making and coping strategies. This approach helps foster adaptive engagement and informed mitigation behaviours without emotionally overwhelming players. Relevant literature has also been incorporated to support these recommendations. This clarification has been added in Discussion section (page 18, line 987-993).

• While you discuss how serious games engage players, you could further explore whether they improve understanding of scientific concepts related to natural hazards. The mention of Bloom's Taxonomy is useful but could be expanded by specifying which cognitive skills these games enhance the most.

**Author response:**

Thank you for this insightful recommendation. In response, we have expanded the discussion of Bloom's Taxonomy to specify which cognitive skills are most enhanced by serious games. The revised text now details how games such as Build a Kit and Disaster Master support foundational levels (Remember and Understand), while Stop Disasters promotes Application, Analysis, Evaluation, and even Creation skills by requiring players to design resilient urban environments and prioritize mitigation

strategies under resource constraints. Furthermore, we explicitly connect these cognitive processes to the understanding of scientific concepts related to natural hazards, such as seismic risk interpretation, flood-prone area identification, and responses to changing weather conditions. This provides clearer evidence of how serious games contribute to both scientific literacy and practical decision-making in realistic risk contexts. This clarification has been added in Discussion section (page 16-17, line 803-880).

• The argument that excessive rewards can be distracting (Chou, 2015) is valid, but it would be helpful to include practical recommendations on how to structure rewards to enhance learning rather than just entertainment.

**Author response:**

Thank you for this valuable recommendation. In response, we have expanded the corresponding section to provide specific and practical guidance on how to structure reward systems to enhance learning rather than solely entertainment. The revised text now emphasizes that rewards should be carefully aligned with core DRM concepts and designed to recognise meaningful progress in addressing realistic challenges. We propose a tiered reward structure that offers progressive recognition, small rewards for mastering foundational content and more significant rewards for demonstrating strategic reasoning and complex decision-making (Boyle et al., 2021). Additionally, we suggest incorporating cooperative rewards to celebrate group accomplishments, which can foster social learning and build collective efficacy (Khalili et al., 2021). This ensures that reward systems support both player engagement and long-term knowledge retention (Ouariachi et al., 2019). This clarification has been added in Discussion section (page 18, line 969-978).

• You mention collective efficacy and community engagement but do not explore in depth how these elements could be effectively integrated into game mechanics. Providing concrete examples from existing games would strengthen this argument Author response:

Thank you for this valuable suggestion. In response, we have expanded the relevant section to include concrete examples illustrating how collective efficacy and community engagement can be effectively integrated into game mechanics. The revised text now describes specific strategies used in existing serious games, such as the assignment of community roles and consensus-building tasks (Tanwattana and Toyoda, 2021), spatial co-design exercises that simulate participatory planning (Olivares-Rodríguez et al., 2022), and time-constrained coordination scenarios requiring distributed leadership (Kano et al., 2016). These examples demonstrate how serious games can foster social resilience by embedding collective efficacy directly into their game mechanics. This clarification has been added in Discussion section (page 17, line 887-898).

**Conclusion**

Reviewer comment:

This section would benefit from deeper reflections on what your research has achieved. The statement that the study provides "new insights" is somewhat vague, be more specific about what is novel in your findings compared to previous research.

**Author response:**

We thank the reviewer for this valuable comment. In response, we have revised the conclusion section to explicitly state the novel contributions of this research and how they advance current knowledge. Specifically, we have clarified the innovative aspects related to the combination of content analysis and expert perspectives, the identification of key educational and communicative features for DRM education, and the comparative evaluation of online games and mobile apps. These revisions ensure that the contributions of our study are clearly differentiated from previous research and provide specific guidance for future developments in this field.

While you acknowledge the limitations of qualitative analysis, it would be useful to suggest how future studies could address these limitations more concretely (e.g., through experimental studies to measure behavioral changes). The recommendations for future research are relevant but somewhat broad. Instead of simply suggesting further studies on different game elements, propose specific research questions or methodologies that could build upon your findings.

**Author response:**

Additionally, we have expanded the limitations and future research sections to provide more concrete proposals. Specifically, we suggest that future studies employ experimental or mixed-method designs to enable causal inference and assess behavioral changes resulting from serious game interventions. We also propose specific research questions to guide future work, including:

- How does narrative tone (e.g., catastrophic vs. non-alarmist) affect players' motivation to adopt DRM-related behaviors?
- In what ways do different feedback and reward structures (e.g., immediate vs. delayed) impact decision-making and knowledge retention?
- How do specific gameplay mechanics (e.g., collaborative vs. competitive strategies or resource discovery) shape the development of problem-solving skills in simulated disaster contexts?

These additions aim to offer clearer directions for future research building directly upon our findings. This clarification has been added in Conclusion section (page 19-20, line 1035-1332).

We hope that the revisions undertaken meet your expectations and we remain at your disposal for any further clarification."

---

## Author Response (AR3)

We sincerely thank Reviewer for the constructive and detailed feedback provided in this round of review. We are grateful for your recognition of the improvements made and for the additional comments that have helped us further refine the manuscript. Below we provide a point-by-point response to each of your suggestions. All page and line numbers refer to the revised version of the manuscript with tracked changes.

**Content:**

**Reviewer comment:**

a) Throughout the manuscript, you keep using different terms for your process with experts ("semi-structured discussions", "online semi-structured interviews", "survey", "expert-led focus groups", "focus group of experts"), which is confusing. In my understanding, you sent out an online survey with open-ended questions, to which 8 experts then replied. If this is correct, please make this clear in throughout the manuscript. You may also add a comment about the rather small size of the expert group in the limitations part of the conclusion and what it means for the reliability of the study.

**Author response:**

Thank you for your observation. We have now homogenized the terminology across the manuscript, consistently referring to an online expert focus group survey to expert. A brief reflection on the sample size and its implications for generalizability has been included in the limitations section of the conclusion (page 19, lines 658-660). Additionally, some sentences that referred to focus group theory have been removed in order to avoid conceptual inconsistencies (page 7).

**Reviewer comment:**

b) Line 117: Why is the reference year 2015? Which hazard has been the most common since then? 2015 was 10 years ago, so saying in the last 20 years floods have been the most common only makes it the most common for half of the timeframe – what about the other half? Add information about what has been the most common hazard since 2015.

**Authors response:**

Thank you for pointing this out. We have removed the outdated reference to the 1995–2015 period and updated the manuscript with recent data from the Natural Disasters Data Book 2023 (ADRC, 2024), which is based on EM-DAT, (the Emergency Events Database maintained by the Centre for Research on the Epidemiology of Disasters (CRED) at the Catholic University of Louvain, Belgium). The revised sentence now states that floods and storms remain the most frequent natural hazards globally, with floods accounting for 44 % and storms 37 % of all recorded disasters in 2023 (page 4, lines 128-130). This update strengthens our analysis by providing up-to-date and specific information on the most recent hazard distribution, responding directly to your suggestion. In addition, the corresponding reference has been added to the References section.

**Reviewer comment:**

c) Figura 1: en los apartados "aplicaciones móviles" y "jugabilidad", indiquen a qué juego se refieren cuando hablan de "sin sistema de recompensas" y "proporciona retroalimentación positiva". En "narrativa" y "juegos en línea", cambien la última línea por: "Sin personajes; el juego se centra en la construcción de edificios o defensas". En "Contenido", dentro del recuadro azul, utilicen una coma después de "términos", es decir: "uso de términos alarmistas, no proporciona...".

**Authors response:**

Thank you for this suggestion. We confirm that the statements "No reward system" and "Positive feedback" apply to all three mobile apps analysed (Geostorm, Disaster Rescue Service, and Earthquake Relief Rescue). We have corrected the corresponding text within Figure 1 to reflect this information. Additionally, this clarification is supported by the detailed descriptions provided in the supplementary material S3 (Table S5). We have also implemented the other requested changes to Figure 1: under "Narrative" and "Online games", the final line now reads "No character; game focuses on constructing

buildings or defences"; and in the "Content" section of the blue box, we have added the comma after "terms" so the phrase now reads "use alarmist terms, do not provide...". All modifications have been applied directly to the Figure 1 to improve accuracy and consistency while preserving visual clarity.

**Reviewer comment:**

Line 283: The European education curriculum comes quite out of the blue – maybe add a sentence or two on what the curriculum states and its aims and why you found it worthwhile to include it in your analysis.

**Authors response:**

Thank you for this observation. We agree that the reference to the European education curriculum required additional context. We have now added two explanatory sentences clarifying the aims of the curriculum and its relevance to the pedagogical goals of serious games in disaster education (page 11, lines 372-374). This improves the coherence of the section and justifies its inclusion in the analysis.

**Grammatical errors:**

• Reviewer comment: Lines 23 to 25 – this sentence is unclear and needs rewriting.

Authors response: Thank you for the suggestion. We have rewritten the sentence (page 1, lines 24–26).

• Reviewer comment: Line 39 – interchange or exchange.

Authors response: We have replaced the word "interchange" with "exchange" as suggested (page 2, line 47).

• *Reviewer comment: Line 53 – could allow the players (eliminate the to).*

Authors response: We have removed the preposition to, resulting in a more concise and grammatically correct sentence: "could allow players to make decisions" (page 2, line 61).

• Reviewer comment: Line 106 – grammatically incorrect, change to: "Considering their characteristics, there is a wide variety of genres and formats such as simulations, which replicate aspects of real or fictional realities and adventures, in which users solve challenges by interacting with people or the environment in a non-confrontational manner."

Authors response: Thank you. We have incorporated your suggested revision (page 4, line 118-120).

• Reviewer comment: Line 126 – exchange "achieve" with "reach".

Authors response: The verb "achieve" has been replaced with "reach" (page 4, line 138).

• Reviewer comment: Line 131 – exchange "destined to" with "intended for"

Authors response: We agree with your observation. The phrase has been revised and we have replaced "destined to" with "intended for" (page 5, line 154).

• Reviewer comment: Line 135 – exchange "DRR is the lack" with "DRR being the lack"

Authors response: We have adjusted the expression to "DRR being the lack" (page 5, line 158).

• Reviewer comment: Line 158 – change to: "A content analysis of the selected games was then carried out. This is a research method used to quantify and analyze the presence, meaning, and relationships of specific words, themes, or concepts, allowing for inferences to be made about the messages within different units of analysis"

Authors response: We have adopted your suggested rephrasing (page 6, line 186-188).

• Reviewer comment: Line 162 – eliminate the "in turn" (i.e. have the sentence go like this: "... tendencies in the games and describes the attitudes ...")

Authors response: The words "in turn" has been removed (page 6, line 190).

• Reviewer comment: Line 164 – change to "These authors adapted the theoretical Social Discourse of Video Games Analysis Model (Pérez-Latorre, 2010) into an analytical instrument for games about climate change using the Delphi method."

Authors response: We have rewritten the sentence as suggested (page 6, line 192-194).

• Reviewer comment: Line 221 – add "could be used in both"

Authors response: We have added the words "could be used in both" (page 8, line 281).

• Reviewer comment: Line 225 – change to: "Finally, in Stop Disasters, decisions such as which buildings to construct or improve, where to build hospitals, or which barriers to build against natural hazards can also involve teamwork."

Authors response: Thank you. We have revised the sentence as suggested (page 8, lines 285-286).

• Reviewer comment: Line 230 – eliminate "in the content"

Authors response: We have removed the words "in the content" (page 8, line 288).

• Reviewer comment: Line 231 – Change to: "Build a Kit is the only game that can likely be completed within an hour, while Disaster Master may also be finished in that time, depending on the students' age and comprehension level."

Authors response: We have incorporated the reworded sentence (page 8, lines 389-390).

• Reviewer comment: Line 234 – eliminate the comma after "actions"

Authors response: The comma after actions has been removed (page 8, line 292).

• Reviewer comment: Line 253 – change to: "As it is based on a fictional film, Geostorm leans more toward fantasy and is less realistic than other games, such as Disaster Master."

Authors response: We have adopted the suggested formulation (page 10, line 340-341).

• Reviewer comment: Line 278 – exchange "however" with "meanwhile"

Authors response: The transition word "however" has been replaced by "meanwhile" (page 11, line 368).

• Reviewer comment: Line 281 – Eliminate the "likewise"

Authors response: We have removed the word "likewise" (page 11, line 371).

• Reviewer comment: Line 299 – change to: "... since natural phenomena lend themselves well to interaction with the environment."

Authors response: We have changed the sentence accordingly (page 11, line 391-392).

• Reviewer comment: Line 318 – I am unsure what that sentence means. If this is what you want to say, please change to: "However, the most prominent element in this context is narrative, since serious games with a strong narrative may have less interaction but can still deliver a powerful impact."

Authors response: Yes, that is exactly the intended meaning. We have replaced the original sentence with the suggested version (page 12, line 418-419).

• Reviewer comment: Line 343 – "since livelihoods, housing, and access to training and information are often unequal and more susceptible to natural hazards"

Authors response: We have modified the sentence as suggested (page 12, line 427-428).

- Reviewer comment: Line 346 change to "training in prevention and vulnerability reduction"

  Authors response: The phrase has been updated (page 12, line 430).
- Reviewer comment: Line 368 change to "integration of findings" o "synthesis of findings"

Authors response: We have adopted the expression "integration of findings" (page 14, line 476).

• Reviewer comment: Line 369 – change to "In this section, the findings of the qualitative content analysis and the expert focus group were compared and further summarized"

Authors response: We have replaced the sentence with the proposed formulation (page 14, line 477).

• Reviewer comment: Line 373 – change to "multiculturalism" instead of "multicultural".

Authors response: We have corrected the term, replacing "multicultural" with "multiculturalism" (page 14, line 481).

• Reviewer comment: Line 468 – there is a word missing

Authors response: Thank you for noticing this. We have identified and corrected the omission to ensure grammatical completeness (page 17, line 580-583).

• Reviewer comment: Line 480 – "Reward system" – eliminate the s in "Rewards"

Authors response: The term has been corrected to reward system, removing the plural "s" as recommended (page 17, line 593).